# Recent Advances and Outlook in 2D Nanomaterial-Based Flame-Retardant PLA Materials

**DOI:** 10.3390/ma16176046

**Published:** 2023-09-02

**Authors:** Lesego Tabea Temane, Jonathan Tersur Orasugh, Suprakas Sinha Ray

**Affiliations:** 1Department of Chemical Sciences, University of Johannesburg, Doorfontein, Johannesburg 2028, South Africa; lmaubane@csir.co.za (L.T.T.); jsurth@gmail.com (J.T.O.); 2Centre for Nanostructures and Advanced Materials, DSI-CSIR Nanotechnology Innovation Centre, Council for Scientific and Industrial Research, Pretoria 0001, South Africa

**Keywords:** 2D nanomaterials, flame-retardants, polymers, nanocomposite, polylactic acid

## Abstract

Poly (lactic acid) or polylactide (PLA) has gained widespread use in many industries and has become a commodity polymer. Its potential as a perfect replacement for petrochemically made plastics has been constrained by its extreme flammability and propensity to flow in a fire. Traditional flame-retardants (FRs), such as organo-halogen chemicals, can be added to PLA without significantly affecting the material’s mechanical properties. However, the restricted usage of these substances causes them to bioaccumulate and endanger plants and animals. Research on PLA flame-retardants has mostly concentrated on organic and inorganic substances for the past few years. Meanwhile, there has been a significant increase in renewed interest in creating environmentally acceptable flame-retardants for PLA to maintain the integrity of the polymer, which is the current trend. This article reviews recent advancements in novel FRs for PLA. The emphasis is on two-dimensional (2D) nanosystems and the composites made from them that have been used to develop PLA nanocomposite (NCP) systems that are flame retarding. The association between FR loadings and efficiency for different FR-PLA systems is also briefly discussed in the paper, as well as their influence on processing and other material attributes. It is unmistakably established from the literature that adding 2D nanoparticles to PLA matrix systems reduces their flammability by forming an intumescent char/carbonized surface layer. This creates a barrier effect that successfully blocks the filtration of volatiles and oxygen, heat and mass transfer, and the release of combustible gases produced during combustion.

## 1. Introduction

Novelty:✓This review presents a complete discussion on 2D nanomaterials as flame-retardants in PLA systems.✓Trends and progress in PLA/2D nanoparticle flame-retardant materials are presented here in a dedicated review paper.✓The challenges researchers face in fabricating and utilizing PLA/2D nanoparticles flame-retardants in PLA systems are presented in a review article.✓The future perspective converses herewith as a guide for emerging researchers in this niche(s).

The extraordinary blend of qualities found in polymer materials, including their lightweight, high specific strength, simplicity of processing, adaptability, cost-effectiveness, and moldability, is driving their rising use in daily life [1]. The search for alternative bio-based/biodegradable polymeric materials has become increasingly important due to the current environmental awareness of the impact of synthetic polymers and the strict environmental laws adopted by governments worldwide [1]. By 2027, the global production of bio-based polymers is expected to expand by an impressive 14% [2]. For instance, the production of bio-based epoxy resin is on the rise, polytrimethylene terephthalate (PTT) has recovered its appeal after several years of stable capacities, and polyethylene (PE) and polypropylene (PP) made from bio-based naphtha are further establishing themselves with rising volumes [2]. The new market and trend report “Bio-based Building Blocks and Polymers—Global Capacities, Production, and Trends 2022–2027” by the International Nova Biopolymer expert group presented capacities and production data for 17 commercially available bio-based polymers in the year 2022 and a forecast to 2027. Polyhydroxyalkanoates (PHAs) and bio-based polyamides (PA) are both expected to see current and future growth. Also returning to the fray is the bio-based PET [2]. The new market and trend report “Bio-based Building Blocks and Polymers—Global Capacities, Production, and Trends 2022–2027” by the International Nova Biopolymer expert group presents capacities and production data for 17 commercially available bio-based polymers in the year 2022 and a forecast to 2027. In 2022, the production was 4.5 million tons, with a total installed capacity of 4.9 million tons, or 1% of the volume of all fossil-based polymers produced. A rise in capacity to 9.3 million tons in 2027 is anticipated, showing an average compound annual growth rate (CAGR) of roughly 14%, much higher than the growth of polymers (3–4%) previously. PHA is predicted to rise by 45%, PLA by 39%, PA by 37%, and PP by 34% until 2027. These polymers will all grow well above the average growth rate. Until 2027, PE use in Europe will rise by 18%, and casein polymers will rise by 15% [2]. After Asia, the leading region with the highest percentage of bio-based polymer(s) production capacity installed globally in 2022 (41%), with the highest capabilities for PLA and PA, Europe follows with 27%, primarily based on starch-containing polymer compounds, PE and PP. South America accounts for 13%, primarily based on PE, and North America accounts for 19%, with significant installed PLA and PTT capabilities. The Australia/Oceania market share of less than 1% is based on starch polymer compounds [2]. 

The focus has been on biodegradable qualities among the countless bio-based polymers, such as starch, methylcellulose, PLA, polycaprolactone, and PHA, as well as their mixtures [1]. Meanwhile, PLA has garnered much interest because of its distinctive qualities, including its attractive look, high mechanical strength, biodegradability, cytocompatibility, transparency, and more [3]. With a projected annual growth of 39% due to these qualities, its applications have expanded [2]. Recent applications of PLA in the automotive sector, industrial carpets, building and construction, furniture, high-end fashion electrical and electronics, items, foams, and fiberfill, among other industries, have sparked its interest in advanced materials [1]. However, PLA is renowned for having a low melting point and a high flammability level, which can occasionally release harmful gases during combustion in contaminated environments. Due to this, expanding the uses for PLA now requires enhancing its FR qualities, even though significant work has already been done in this area [4,5,6,7].

Like any other polymer, the flammability of PLA polymer is determined by well-known factors like solid degradation rate and/or peak heat release rate “pHRR” (burning rate), delay time, ignition temperature, critical heat flux for ignition (ignition characteristics), in particular toxic species emissions (product distribution), production of smoke [6], and others. Previously, the only FRs for PLA that were conventionally used were those with halogen moieties because they have been largely successful in most polymers [1]. However, due to their adverse health effects on both humans and animals, these FRs are now recognized as major potential global contaminants [1]. As the dust from indoor exposure, mechanical recycling of plastics and metals, and open burning and cremation of household items, halogenated FR leaks into the environment throughout their valuable lives. Throughout the lifespan of treated objects, halogenated FR leaches to the environment as dust through indoor exposure, mechanical recycling of plastics and metals, and cremation and open burning of household trash, such as electronic components, paints, solvents, and textiles [1]. They are also mainly linked to endocrine and thyroid disturbance, immunological toxicity, reproductive problems, cancer, neonatal and fetal defects, abnormal child development, and neurologic functions [1]. Due to rigorous government regulations and increased consumer awareness of the environment, halogenated FRs are consequently being phased out gradually [1]. The research community has produced many intriguing findings in response to the search for substitute eco-friendly FRs. However, no commercially available two-dimensional (2D)-based FR has been specifically developed to manage the flammability of PLA to retain its biological integrity while enhancing or maintaining its delicate crystalline and mechanical qualities [1]. Though the literature has presented reviews on 2D layered materials as FRs [8], no review article or book is dedicated to or focused on PLA-based FR NCPs affected by including 2D nanomaterials as the active compounds/additives. We have covered this existing gap in this paper. This review mainly focuses on additives other than pure bio-based systems. It discusses recent developments in the 2D FR PLA additives [3,9]. In addition, the review sheds light on 2D nanoparticle (NP)-based FR loading and its relationship to critical fire safety indicators, including limiting oxygen index (LOI), vertical burning test (UL-94), pHRR, and others. We critically discussed the importance of FR additives, such as graphene, MXene, montmorillonites (MMTs), and fillers and their role as advanced FR nanofillers in PLA composites.

### 1.1. PLA Feedstock

Lactic acid (LA) molecules create biodegradable and biobased PLA. Technically, PLA is suitable for various applications, from single-use packaging to lasting consumer items, because of its flexibility and other technical characteristics. The feedstocks used are glucose and sucrose. The primary biological source of glucose is still corn grains. The grains of maize are processed to remove the starch after harvest. Then, a process known as hydrolysis converts starch molecules into molecules of glucose. After being dried, glucose goes through an industrial fermentation process using bacteria. The resultant LA solutions are transformed into lactide and then crystallized to remove impurities before polymerizing into PLA. Similar general steps apply when using sugar cane, except the fermentation process uses sucrose extracted from the stalks [10]. Because scientific expertise and production capabilities have risen relatively high, PLA manufacture is currently more economical than other bioplastics. PLA has the greatest biopolymer production capacity installed as of 2022 in the global production capacities of all biopolymers, with a reported 39% growth rate in market shares [2]. Again, the European Bioplastics report on bioplastic production shows that PLA production stands out at 20.7% higher than all others and is predicted to rise to 37.9% by 2027 [11]. The three main application areas for PLA-based finished goods are rigid packaging, flexible packaging, & textiles.

The main goal of PLA feedstocks/synthesis processes has been to address the environmental concerns related to land usage and competition with food products tied to the current feedstock creation of food crops. Among these, the use of leftover plant material from their cultivation (stover) and processing (e.g., sugar cane bagasse) as cellulose-based feedstocks has also been studied for a while. These materials are obtained as a byproduct, eliminating or at least lessening first-generation feedstocks’ drawbacks. Several recommendations have been made to entirely uncouple PLA manufacturing from agricultural land usage in recent years. Waste and byproducts from the food business that would otherwise have little or no economic value are also used nowadays. Harbec [12] and Broeren et al. [13] have reportedly examined water waste generated during the industrial preparation of potatoes as feedstock for PLA synthesis. Using lactose and proteins as feedstocks, Liu et al. [12] look into manufacturing LA from cheese whey by contrasting various bacteria species and fermentation enzymes. Juodeikiene et al. [13] investigated how to increase the yield from cheese whey. Nguyen et al. [14] investigated an instance in which waste from the industrial extraction of curcuminoid from the Curcuma longa root utilized in medical applications is fermented to LA through simultaneous saccharification and fermentation. According to De la Torre et al. [15], orange peel waste and corn steep liquor have also been used as substrates. Coffee pulp, a byproduct of coffee processing, is the subject of Pleissner et al. [16] investigation. Alves de Oliveira et al. [17] suggested using sugar beet pulp as a byproduct of sugar extraction from sugar beets for animal feed. Another area looks at the possibility of switching from land-based to sea-based resources. It is thought that the cultivation and fermentation of carbohydrate-rich marine plants will provide a chance to create whole new production pathways, protecting current food supply chains from disruption by the creation of plastic. In this vein, Helmes and his group [18] investigate the utilization of the seaweed *Ulva* spp. in the LA synthesis process. Brown algae species Laminaria sp. have been reportedly grown as a feedstock source in an experiment by Ögmundarson et al. [19].

The number of novel feedstocks for PLA for which thorough studies on environmental performance are available is still relatively small. Lignocellulose, a second-generation feedstock produced as a byproduct of plant cultivation, has drawn particular interest in the already available data. Its primary benefits are clear: no environmental concerns are connected to land transformation when gathered only as a byproduct within the framework of currently used production processes for corn, sugar, or cereal. Additionally, the allocation principle’s emission reduction also applies to the cultivation of the feedstock since fertilizer input, water consumption, and machinery used are effectively shared between the agricultural outputs. By doing this, it would be possible to lessen the effects of climate change on both local and global environmental categories, including those linked to soil, water, and human health. Although these agricultural by-products are frequently not put to significant commercial use, they are not useless from an ecological standpoint, as shown by Ögmundarson et al. [19], and this should be taken into account in an environmental study. The plant material left on the field after harvesting offers ecological benefits, including erosion control and a boost to the soil’s carbon content, or it can be burned to provide electricity instead of using fossil fuels. The potential costs of these alternate uses should be considered in LCAs, examining the advantages of cellulose feedstocks, ideally through system expansion. Such a consideration can significantly degrade the overall environmental performance, according to the analysis by Ögmundarson et al. [19].

In contrast, organic waste undergoing industrial processing appears to be better because there aren’t many other choices. Even more so, their usage as a feedstock for bioplastics prevents a potential final disposal that could be emission-intensive in landfills or the environment. The emission of the carbon included in these particles as CO_2_ back into the environment is at least postponed for a potentially long period by integrating it into polymeric materials. But there are also unique environmental dangers associated with this group of feedstocks, most of which are brought on by the processing technology’s immaturity. The environment’s delicate equilibrium is significantly harmed by the bio-refinery process’s existing high energy intensity. Utilizing algae and allied sea plants, another novel category of PLA feedstock in the literature (Table 1), bears a similar caution. The processing’s complexity and high energy requirements also provide an environmental burden. Future efficiency improvements due to scale effects, process innovation, and increased renewable energy percentages in the energy mix of the producing nations may be able to address this issue. To what degree this will decrease emissions in the medium term is still debatable.

### 1.2. PLA Synthesis in Brief

The category of polymeric biomaterials referred to as poly-α-esters, poly-α-hydroxy acids, and/or aliphatic polyesters includes PLA, a hydrophobic polymer. It is made from LA, also known as 2-hydroxy propanoic acid, which has two enantiomeric forms, L-(+)-LA and D-(-)-LA and is a water-soluble monomer. LA is the starting material for this process, as indicated in Figure 1 [10].

Even though both enantiomers are often used in industrialized synthesis, L-(+)-LA is its isomer of interest for biotechnology, bioengineering, and biomechanics applications, seeing it participates in the metabolism at a cellular level within the body system and lowers the likelihood of adverse reactions. L-(+)-LA could be excluded from the body in the form of H_2_O and/or CO_2_ from the lungs in an in vivo setting by either incorporating it into the Krebs’ cycle or converting it into its stored form in the liver “glycogen” [20]. If L- and D-monomers racemic blend is utilized/adopted, poly-D,L-lactic acid (PDLLA) copolymer is generated. PLA could then be made by commencing with virgin L- and D-lactic isomers, corresponding to poly-L-lactic acid (PLLA) and poly-D-lactic acid (PDLA) homopolymers. Stereochemistry significantly impacts material properties: PLLA is a semi-crystalline polymer, whereas PDLLA is an amorphous polymeric material with no melting point. Additionally, because PLLA has crystalline areas, its degradation rate is considerably slower than PDLLA’s. Regarding the production of LA, several methods can be used; the most common one is as follows [21]:(1)$${\mathrm{CH}}_{3}\mathrm{CHO}+\mathrm{HCN}\to {\mathrm{CH}}_{3}\mathrm{CH}\left(\mathrm{OH}\right)\mathrm{CN}$$
(2)$${\mathrm{CH}}_{3}\mathrm{CH}\left(\mathrm{OH}\right)\mathrm{CN}+2H_{2}O+\mathrm{HCl}\to {\mathrm{CH}}_{3}\mathrm{CH}\left(\mathrm{OH}\right)\mathrm{COOH}+{\mathrm{NH}}_{4}\mathrm{Cl}$$
(3)$${\mathrm{CH}}_{3}\mathrm{CH}\left(\mathrm{OH}\right)\mathrm{COOH}+{\mathrm{CH}}_{3}\mathrm{OH}↔H_{2}O+{\mathrm{CH}}_{3}\mathrm{CH}\left(\mathrm{OH}\right){\mathrm{COOCH}}_{3}$$

When lactonitrile is produced from hydrogen cyanide and acetaldehyde (1), it is hydrolyzed at a low pH to produce LA (2), which is then transformed to methyl lactate (3) via esterification and finally reclaimed, followed by its purification via distillation. After hydrolyzing, methanol is recycled in step two to create LA and methanol from lactate (3). In any case, a racemic mixture results from this kinetic process [21].

The most popular method for processing sugar solutions at the moment is bacterial fermentation; this method produces high yields and, depending on the bacteria used, permits the production of either a specific stereoisomer or a racemic mixture. According to estimates, this method yields around 90% of global LA today. The following LA purification procedure, which is costly and affects process profitability, is the crucial stage in this system. Reactive distillation, membrane separation, ion exchange, and liquid extraction are frequently used.

Step growth polymerization or ring-opening polymerization (ROP) are viable methods for creating PLA polymers. The reactivity of the two LA functional groups is exploited in step growth polymerization; in fact, the polycondensation of the hydroxyl and carboxyl moieties results in the production of the ester linkages that make up the polymer backbone. This method of synthesis has several drawbacks, including the need for prolonged residence times for longer chains (which can cause undesirable side reactions, such as transesterification), difficult reaction parameters with temperatures reaching 250 °C, and a vacuum pressure of ~100 mbar), along with the continuous exclusion of water (a polycondensation byproduct). It is theoretically possible to use chain extenders (such as isocyanates or epoxides), but doing so inevitably affects the quality and purity of the material.

ROP is the most used method at the industrial scale because of its benefits, mild process conditions, brief residence durations, the absence of side products, and high molecular weights. The most extensively used catalyst in PLA synthesis is 2-ethylhexanoic tin(II) salt, likewise known as stannous octoate “Sn(Oct)_2_”, which has been given U.S. Food and Drug Administration approval and is typically used in conjunction with alcohol as a co-catalyst. The availability of cyclic monomers and their optical and chemical purity are the primary bottlenecks of ROP because impurities have a negative impact on material quality due to the reaction’s sensitivity to remaining non-cyclic monomers. The cyclic dimer lactide, having trio-stereoisomeric forms and serves as PLA raw material, is represented in Figure 2 as the following: D, L-, LL-, and DD- (also referred to as a meso-lactide) [10,21]. 

A backbiting kinetic mechanism Is typically used to create lactide, starting with a low MW pre-polymer. After distillation, the cycles are eventually collected. Enzymatic polymerization and azeotropic dehydration are further methods of synthesis. The routes for synthesizing PLA-based polymers are listed in Figure 2 [22].

Due to its eco-friendliness, biocompatibility, and processability, PLA is now extensively used in materials science, even in applications where retarded flammability is required. PLA is processable using a variety of methods, including melt compounding and extrusion, injection stretch blow molding, injection molding, solution intercalation plus casting, blow film extrusion, foaming, melt spinning of fibers, electrospinning, 3D printing, etc. [3,23,24,25].

It is also important to note the advantages of PLA as a polymer: its eco-friendliness, biocompatibility, ease of processing, and less energy consumption (25–55% less energy) during production compared to petroleum-based polymers. Though several advantages have been stated, PLA has its disadvantages, which are poor toughness (highly brittle: <10 elongation at break), slow crystallization and degradation rate, relatively high hydrophobicity (contact angle of 80°), and a lack of reactive side chains or surface functional groups making it very difficult to almost impossible to modify PLA.

### 1.3. Polymeric Materials Behavior in the Fire: PLA Exacts

Before tackling the issue of the fire threat provided by polymeric materials, it is critical to gain a complete understanding of the combustion process of a polymer. Polymer combustion is a multi-step process involving several related chemical and physical steps. Three crucial components, heat, fuel, and oxygen, sometimes shown as a fire triangle, must all come together simultaneously for polymer combustion to begin and continue [26]. Due to their chemical composition, which is mainly composed of carbon and hydrogen, polymers are highly combustible [27].

A combustible substance (reducing agent) and a combustive substance are required for the combustion reaction (oxidizing agent). Combustion is usually brought on by oxygen in the air. When a heat source raises the temperature of the polymeric material to a threshold where polymer bonds split, the entire process frequently starts there. A combustible gaseous mixture is created when the polymer fragments are created due to the volatile portion diffusing into there (also called fuel). This gaseous mixture ignites and releases heat when the auto-ignition temperature, which is the temperature at which the activation energy of the combustion reaction is achieved, is reached. The fuel can ignite when exposed to a potent energy source outside of it at a lower temperature (referred to as the flash point) (spark, flame, etc.). How long the combustion cycle lasts depends on how much heat is produced during fuel combustion. When the amount of heat released exceeds a certain limit, new breakdown reactions in the solid phase start, forming more combustibles. Consequently, the combustion cycle is preserved, and the phrase “fire triangle” is utilized (Figure 3) [27].

The physical characteristics of the material determine how much energy is required to initiate combustion in polymers. Semi-crystalline thermoplastics, for instance, cause the polymer to soften, melt, and drip when heated. The polymer’s capacity for heat storage, as well as its enthalpy of fusion and crystallinity level, all play a role in determining how much energy it can hold throughout these processes. The exothermicity of the processes, the specific heat, and the thermal conductivity of the semi-crystalline thermoplastic all contribute significantly to the temperature increase of the polymer and the accompanying rate. However, most thermosets and amorphous thermoplastics experience polymer breakage during heating because they lack a melting point [27]. 

The endothermic process of covalent bond dissolution, which results in the thermal breakdown of a polymer, requires energy input. The energy introduced into the system must be greater than necessary to bind the covalently linked atoms together (200–400 kJ/mol for the bulk of C–C polymers). The decomposition mechanism’s major factors include the weakest bonds and whether oxygen is present in the solid or gas phases. The combined effects of heat and oxygen typically bring on thermal breakdown. As a result, oxidizing and non-oxidizing heat degradation can be separated [28].

#### 1.3.1. Principle of Polymeric Materials That Resist Fire

The pyrolysis and combustion of polymers occur in several phases, as shown in Figure 3. The polymeric substrate, which an outside heat source has heated, is converted into combustible fuel by pyrolysis. Only a part of this fuel is normally completely consumed when mixed with the stoichiometric amount of oxygen in the flame. Strong catalysts and abundant oxygen, among other severe techniques, can burn up the residual fraction. The substrate keeps pyrolyzing as the combustion cycle is maintained by returning some of the heated energy. Additional heat is removed by the environment [29]. Depending on their makeup, FR systems can function chemically or physically (by chilling, forming a protection barrier, or diluting fuel) (reaction in the condensed or gas phase). They might prevent different polymer combustion processes from happening (heating, pyrolysis, ignition, propagation of thermal degradation) [27].

##### Physical Action

Some FR additives experience an endothermic breakdown, which causes heat to be consumed and the temperature to drop. This necessitates a small amount of reaction medium cooling to below the temperature of polymer combustion. Magnesium hydroxide and hydrated trialumina also belong to this category and start to produce water vapor at temperatures of about 200 and 300 °C, respectively. This kind of strong endothermic reaction is said to act as a “heat sink”. The breakdown of the FRs dilutes the combination of flammable gases. It produces inert gases like water, carbon dioxide, etc., which lowers the concentration of reagents and the risk of ignition. Various FR additions may also form a protective solid or gaseous layer between the solid phase, where thermal degradation occurs, and the gaseous phase, where combustion occurs. The passage of such a barrier prevents gases like oxygen and flammable, volatile vapors. As a result, significantly less decomposition gas is produced. Additionally, the fuel gases and oxygen can be physically separated, halting the combustion process [27].

##### Chemical Action

The amount and composition of the gaseous products and the energy needed to heat the polymer to the pyrolysis temperature and break down, gasify, or volatilize the combustibles all affect how flammable the substrate is. An FR with a condensed phase chemical mechanism can alter the pyrolytic route of the substrate and considerably reduce the number of gaseous combustibles by encouraging the formation of carbonaceous char and water. In this case, as the flame-retarding chemical concentration is increased, the heat released during combustion is decreased [29]. 

The heat released during combustion decreases typically as the flame-retarding chemical grows, although the amount of combustible material in the gas-phase mechanism doesn’t change. The pyrolysis decreases or stops when the surface temperature decreases because less heat is returned to the polymer surface. When it comes into contact with the flame, the flame-retarding substance must be gaseous and volatile. Instead, it must disassemble and deliver its molecule’s active component to the gaseous phase. The less active agent will be present in the char that remains in the substrate. In the worst situation, the polymer pyrolysis should proceed as if no FR had been added. As they approach the flame, the composition of the volatiles should also not be affected by the presence of the gas-phase active agent [29].

#### 1.3.2. The Gas-Phase Mechanisms

Based on its capacity to prevent the polymer’s combustion, the active FR possesses gas-phase activity. Similar to how other fuels do it, polymers also go through pyrolysis to create species that can react with airborne oxygen, giving rise to the branching H_2_-O_2_ scheme that fuel combustion uses as its fuel [29]:(4)$$H^{{}^{\circ}}+O_{2}={\mathrm{OH}}^{{}^{\circ}}+O^{{}^{\circ}}$$
(5)$$O^{{}^{\circ}}+H_{2}={\mathrm{OH}}^{{}^{\circ}}+H^{{}^{\circ}}$$

The primary exothermic process that provides the bulk of the energy required to maintain the flame is:(6)$${\mathrm{OH}}^{{}^{\circ}}+\mathrm{CO}={\mathrm{CO}}_{2}+H^{{}^{\circ}}$$

The chain branching reactions (4) and (5) must be inhibited for the combustion to be slowed or prevented in polymeric systems like PLA. 

#### 1.3.3. The Condensed-Phase Mechanism

The condensed-phase method relies on the idea that the polymer and the FR component, frequently provided in large amounts, would interact chemically. The pyrolytic breakdown occurs at higher temperatures than this contact. It was suggested that the two main types of this interaction were cross-linking and dehydration. They are well-established for various polymers, including synthetic and cellulosic ones [29].

#### 1.3.4. FR Mechanism of PLA-2D NPs NCP in Brief

There is no one method for determining flame retardancy because the FR chemistry of composites is typically overly complex. For this reason, scientists frequently select FRs based on models for the chemistry of heat degradation of polymers and fire hazards. In addition, various commercial specifications, including those relating to the way a product is processed, its cost, environmental stability, color, and, more recently, its sustainability and recyclable nature, are imposed on commercial goods.

It is commonly acknowledged that the basic FR mechanism of polymeric NCPs, such as PLA systems made with 2D layered nanomaterials, is based on those materials’ barrier characteristics. Furthermore, the addition of layered nanomaterials can operate as a carbon donor (charring agent) to increase the char residue, forming an insulating char layer between the burned and unburnt structures during the combustion process [8]. For instance, it has been shown that layered double hydroxides (LDH) increase the PLA’s flame retardancy by leaving behind refractory oxide residues on the surface of the material and releasing water vapor and carbon dioxide during the polymer’s breakdown [30].

Through a 2D layered structure with a lamella blocking effect, graphene can have numerous special properties that limit oxygen access and delay the heat transfer between interfaces, thwart the escape of pyrolysis products, and mix the oxygen [31]. Again, it has been hypothesized that MMT creates a carbonaceous silicate during combustion on the surface of the polymer material. Heat and mass transport are inhibited by the carbonaceous silicate [32]. The intercalated particles under the heat source can expand perpendicular to the carbon layers in the crystal structure because expanded graphite (EG), one type of graphite intercalation compound, retards flame in its PLA systems. These structures are intumescent and can rely on thermally induced disintegration to create a char layer that separates the substrate from the heat source [33]. It has also been claimed that the dispersion and exfoliation of molybdenum disulfide (MoS_2_) in PLA provide a physical barrier effect that can impede the transfer of heat and the products of polymer decomposition. To limit the transmission of heat and mass during combustion, the transition metal element Mo encourages the formation of a physical barrier known as the char layer [34]. Layered materials slow down hazardous gas escape from matrices into the surface of PLA-2D NPs NCPs, reducing the amount of flammable gases in combustion zones. The “labyrinth effect” is another name for this.

Furthermore, incorporating layered nanomaterials increases the viscosity of polymer matrices, which may help slow down oxygen diffusion rates inside polymer melt and the generation of breakdown products from below. As the temperature rises during pyrolysis and combustion, the viscosity and density of the polymer melt also rise. As a result, the mass loss rate (MLR) gradually decreases, and the movement of FR particles to the flame slows down. In addition, the higher the viscosity of the polymer melts, the lower the likelihood of dripping and the greater the likelihood of forming an expanding layer of char [35].

## 2. Processing of PLA-2D NPs NCPs

Aimed at the preparation of FR PLA-2D NPs NCPs systems, researchers around the globe have majorly adopted processing approaches, including melt processing/compounding (MCD), in situ polymerization, solution intercalation/casting, electrospinning, melt spinning, injection molding, and others [1,4,5,10,17,23]. This section discusses these approaches, with literature references.

### 2.1. MCD/Intercalation

The best method for creating thermoplastic and elastomeric polymeric matrix-based 2D NPs/polymer NCPs is MCD. Usually, a banbury or an extruder is used to melt the polymer and mix it with the necessary quantity of intercalated clay. An inert gas, such as argon, nitrogen, or neon, is used to mix melts during the process. An alternative method is to dry mix the polymer and the intercalant, heat the mixture, and apply enough shear to the mixture to create the required 2D NPs polymer NCPs. Compared to polymer solution intercalation or in situ intercalative polymerization, MCD has many advantages. This approach combines the NPs and PLA melt during the melting process. The procedure is carried out under shear, usually at a temperature higher than the melting point of the PLA. This method offers several benefits compared to solution casting and in situ intercalative polymerization processes. First, since no solvent is used, the method is environmentally benign. Second, this method is compatible with polymer industry-used processing methods like extrusion and injection molding. Several instances where MCD has been adopted for processing PLA/2D NPs FR NCP systems are given later in this article.

MCD is compatible with modern industrial procedures like IM and extrusion. The MCD method has gained popularity due to its potential in industrial applications.

MCD involves producing plastic materials incorporating 2D NP formulations by mixing and/or blending the polymers and NPs in a molten state to achieve the desired features. This is done to fabricate polymer systems that are flame-resistant. These mixtures are often automatically dosed through feeders or hoppers with predefined setpoints.

In a particular instance where MCD of 2D NPs reinforced FR polymer NCP system is presented, a group of researchers synthesized MXene (Ti_3_C_2_) by selective etching of Ti_3_AlC_2_ with LiF/HCL solution to obtain Ti_3_C_2_–(OH). Afterward, the Ti_3_C_2_–(OH) was treated (selectively etched) with DMSO by ultrasonication, followed by washing and drying and then inclusion in PLA matric via MCD, as shown in Figure 4. Before processing, PLA and fillers underwent a 4-h vacuum drying at 80 °C. The mixtures were created by combining PLA granules and Ti_3_C_2_-DOPO powders (0.5 and 3.0 wt.%) in a twin-screw extruder operating at 170 °C, 20 g/min feed rate and 120 rpm screw speed. In conclusion, the extrudates were cooled in a water bath, divided into pellets, and vacuum-dried at 60 °C for 24 h [36]. 

In another report, Yu et al. [37] presented the successful utilization of the MCD approach to fabricate FR PLA/GO-x/CSFR hybrid system, as shown in Figure 5.

### 2.2. In Situ Polymerization

This is called in situ polymerization when polymer chains grow in the same environment as nanomaterials. In situ polymerization entails first pre-blending the NPs with the monomer/monomer solution with heat, radiation, or appropriate initiators. Diverse research groups have reported successfully adopting this approach to synthesize PLA-2D NPs FR NCP materials. This approach has been adopted by researchers either on its own or in conjunction with other approaches because it provides a polymer NCP system having better interfacial interaction between the polymer and the active FR 2D NP, reduced or eliminated 2D NPs self-aggregation tendencies, uniform distribution of the 2D NPs withing the host matrix, reduced processing steps, along with several other benefits not mentioned here.

### 2.3. Solution Intercalation/Casting

This technique combines a solution of PLA in a suitable solvent with a dispersion of the 2D NPs in the same or a different solvent. For instance, when creating PLA NCPs using clay, the solvent is typically used to pre-swell the clay before the dispersion of the swelled clay is combined with the PLA solution. The PLA chains intercalate, displacing the solvent from the 2D NPs (such as nanoclay (NC)) galleries and absorbing onto its surface. The catalyst for this approach is the increase in entropy caused by the desorption of the solvent molecules, which makes up for the loss of entropy brought on by chain confinement in the intercalated structure. A PLA composite/hybrid system having an intercalated structure is created due to the solvent’s evaporation. Several researchers have also employed this approach to create PLA-2D materials for various purposes.

### 2.4. Electrospinning

Electrospinning was used to create the first nanofibers more than four centuries ago. William Gilbert’s work from 1600 was seen as an early example of what would later develop into the contemporary electrospinning technology [38]. Electrospinning is undoubtedly the most well-studied top-down technique for creating nanofibers from various organic and inorganic materials. It is well-known and talked about. ES is a dry spinning technique that uses electric force to draw fibers from a liquid polymer solution or melt [39]. The fiber production process from the liquid is physical, either via solvent loss or freezing of the melt. The procedure has recently become popular in the lab for creating continuous nano-scale fibers. Additionally, well-established industries use electrospinning to create highly effective filters.

This approach has been adopted richly to fabricate diverse FR advanced materials. Some researchers have also used it to prepare PLA-2D materials aimed at diverse applications. 

### 2.5. Melt Spinning

In this approach, the 2D NPs are combined with the PLA melt during the melting process. The process is usually carried out under shear at a temperature over the melting point of the PLA. This approach has many advantages compared to solution casting and in situ intercalative polymerization methods. The procedure is environmentally benign because no solvent is needed in the first place. Second, the processing methods employed in the polymer industry, such as extrusion and injection molding, are compatible with this procedure.

PLA molecular chains' thermodynamic interaction with the reinforcing NPs determines how 2D NP-based PLA NCPs are formed during melt processing. Regarding NCs, it is crucial that PLA chains spread across NC’s silicate galleries [3,5,10]. The level of dispersion obtained during the melt processing stage relies upon the enthalpic interaction between the PLA chains and the NPs and the processing circumstances. Micro-composites occur when a nanolevel of dispersion is impossible due to unfavorable enthalpic interactions. The improvement of processing conditions is equally significant. The mechanism of intercalation/exfoliation is more of a shearing process in the formation of polymer/2D nanoplatelet clay composites, in which the platelets of the NC tactoids are peeled apart/separated by a combination of mechanical and chemical forces, resulting in the formation of smaller tactoids. This notion departs from the accepted theory [40], which states that the polymer chains enter the clay and increase the d-spacing between the platelets to overcome the van der Waals forces and cause their separation. Other research teams have also proposed the peeling mechanism and the significance of shear in developing polymer/clay NCPs [41]. To achieve a greater amount of delamination, Treece et al. [42] and Zhu and Xanthos have emphasized the significance of both a high shear rate and a longer residence time [43]. Additionally, according to Bandyopadhyay et al. [41], residence time and feed rate have a power–law relationship, with a decline in feed rate leading to an increase in residence time. Similar trends have also been documented by Poulesquen and Vergnes [44] and Bigio et al. [45].

A relatively new technology, such as processing with supercritical carbon dioxide [46,47], has gained importance in manufacturing PLA NCPs in addition to the methods already discussed. Particularly since this solvent is more environmentally friendly than other organic solvents, processing with supercritical CO_2_ as a medium while in situ polymerization [46] or melt processing [47] has attracted interest. It has been discovered that supercritical carbon dioxide works well to widen the interlayer gap and, as a result, aid in dispersion when clay serves as the nanofiller. This method, however, is not appropriate for dispersing other NPs, such as carbon nanotubes (CNTs), graphene, or nanocellulose, within a PLA matrix.

It is important to remember that the matrix has reportedly degraded when PLA is processed in a melt. When alkyl ammonium-based surfactants are employed to modify the surface of NCs, this propensity is exacerbated in the presence of organically changed NCs. Hoffman elimination reaction is initiated by alkyl ammonium surfactants at about 200 °C, which could trigger PLA matrix degradation during processing. Researchers have developed NCPs using modified NC surfaces modified with phosphonium salt to remove or balance out major differences in the thermal degradation characteristics of PLA NCP during melt processing [48].

## 3. PLA/2D NPs FR Materials’ Trend and Advances

The available and application-proven FRs that have been conventionally used in plastic products available in the market are metal oxides, phosphorous and its compounds (e.g., Organophosphate FRs (OPFRs)), halogenated compounds (e.g., Hexabromocyclododecane (HBCD), Tetrabromobisphenol A (TBBPA)), melamines (e.g., melamine cyanurate), sulfur-containing compounds (e.g., Poly[sulfonyl(bis-4-phenyl)phenylphosphonate], diammonium imidobisulfonate, fluorinated sulfonamides, etc.), metal hydroxides (such as., magnesium hydroxide), etc. amongst which in most cases are highly toxic and have been banned by governments/regulatory bodies around the globe.

Macro/microsized FR compounds are mostly adopted as active FR additives in polymeric systems. One of the most notable downsides of macro-sized/micro-sized FR agents that are conventionally used is their high loading rate of 40–70% in the composite material, adversely affecting composite mechanical properties. A nano FR agent spread exists in one of the phases of PLA-2D NPs composite material. Literature has demonstrated that adding 1 (10 wt.% of 2D NP-based FRs in PLA systems can significantly slow the heat release rate, delay ignition, and lower the flame propagation rate.

The cutting-edge trend of creating biodegradable materials is becoming increasingly popular to preserve our natural resources and leave a legacy for future generations. The most widely used biodegradable composite materials have a polymer matrix and natural fiber reinforcing filler, and they can lose at least 90% of their initial weight within six months of use. These biodegradable materials have many applications and are most frequently used in construction and automotive industries. The user should always be at the forefront of the designer’s mind while creating such materials, and the emphasis should be placed on their health and safety. Preparing for fire and averting potential fatalities are two of the most crucial safety considerations. Considerable heat and smoke emissions during a fire could seriously harm people and destroy many of their possessions. It is crucial to direct the design of biopolymer-based composites to match flame-retardant performance with environmental standards because a sizable portion of the products available on the open market today fall into the category of composites. The strict standards imposed on the automotive and construction industries require that traditional composites be amended through various changes.

To overcome the burning shortcoming and increase the application of commonly used PLA-based composites, they must be treated with FR materials on some or all of their constituent parts. The use of FR 2D NPs to successfully halt/retard the burning process (heating, breakdown, ignition, combustion, and flame propagation) of PLA composites has recently attracted much attention as per the literature. There is a significant desire for FR nanofillers to be eco-friendly, considering the future vision, described with terms like green, eco, sustainable, and so forth. A new type of composite known as NCPs has evolved as an alternative to traditional non-biodegradable petroleum-based plastic materials. This has made it possible to use advanced, high-performance, lightweight green NCPs.

The Green Plan’s achievement, which calls for only environmentally friendly substances and zero waste production, is the main focus of current concerns regarding creating biodegradable FR NCP systems.

The niche of PLA-2D NP-based FR NCPs has steadily grown within the last decade, as depicted in Figure 6. The growth in this area is primarily related to the global drive towards green materials for which FR materials, such as PLA-2D NPs systems, are not exempted.

Though initially, researchers and industries broadly utilized non-ecofriendly FRs in polymeric matrices that were also not eco-friendly, the current trend is the use of 2D NPs, such as MXenes, graphene and its derivatives, NCs, LDHs, etc., which are relatively environmentally friendly and well as eco-friendly/biopolymers, such as PLA which is the focus in this article.

An excellent example in this regard is the work presented by Guo et al. [49], where the authors utilized 2D Cloisite 30B (C30B) in synergy with melamine polyphosphate (MPP) in PLA FR fabricated NCP systems. Though the inclusion of 17% MPP enhanced the flame retardancy of the PLA matrix, leading to the UL-94 rating of V-2, there was a drastic decline in the mechanical property performance of the formulated. However, upon the inclusion of 1% C30B, the reduced mechanical performance was reversed, resulting in enhanced mechanical performance as well as flame retardancy: a UL-94 rating of V-0 was obtained in the composite systems having just 1% C30B. The preparation of the NCP was achieved using a Fused Deposition Modeling (FDM) 3D printer and compression molding (CM) to compare the final products' property performance [49]. Additionally, the reported LOI value decreased from 33% for the virgin PLA to 27% for the sample having 1% C30B (P17M1C) and a corresponding pHRR reduction from 342 for the virgin PLA to 198 for P17M1C.

## 4. 2D Flame Retarding NPs as FRs in PLA Composite Systems

Several layered NPs (2D nanomaterials), such as expandable graphite (EG) [50], MXenes [36], MMT [51], LDH [52], MoS_2_ [34], and others, have so far demonstrated outstanding flame-retarding properties in PLA-based polymer composite systems.

It has been suggested that the geometry of NPs, such as zero-dimensional (0D), one-dimensional (1D), 2D, and/or three-dimensional 3D, greatly affects the FR properties of polymeric matrices and their ultimate NCP(s) [53]. It has been demonstrated that fire performance increases with the order of rod-like, spherical, and plate-like geometries that qualitatively match the effective surface area of NPs in the PLA-based NCP(s). This phenomenon was claimed to be dependent on the intercalation/exfoliation (nanodispersion) of the clay (MMT), which resulted in a significant increase in the surface area that aided in the rapid migration and accumulation of platelets on the exposed sample surface before the formation of intumescent char. A thicker aluminum phosphate/MMT NCP char formed as a result, acting as a potent transport barrier and preventing intumescence [53]. Additionally, investigations into the microstructural development of the leftovers revealed remarkably uniform, hollow-fibrillar formations after PLA/aluminum diethylphosphinate (AlPi) combustion.

### 4.1. Nanoclays (NCs)

The most widely used and significant class of two-dimensional NPs are often NCs (amongst which a notable exception of increasing importance is one-dimensional (1D) sepiolite). Most of them are phyllosilicate minerals that are found in nature. Although boehmite (an aluminosilicate) and other clays have also been utilized in polymeric (nano)composites, MMT is by far the least expensive and most frequently employed [54]. Even compared to other nanomaterials, they compare favorably in cost-benefit ratio [55].

2D Layered mineral silicate NPs called NCs have been developed for usage in various niches. Due to NC's potential advantages, including enhanced mechanical strength, lower gas permeability, and superior flame-resistance [6], polymer-clay NCPs are a class of nanomaterials (NMTs) that researchers have widely explored. Based on their chemical makeup and NP form, NCs are classified into various classes, including MMT, halloysites, bentonite, kaolinite, and hectorite [3,48,56]. The alluring class of hybrid organic-inorganic NPs known as “organoclays” can be used in polymer NCPs as FRs, gas absorbents, drug delivery vehicles, and rheological modifiers [6]. Along with the well-known FR chemistries, it has been extensively reported that the application of NC enhances the qualities of PLA and its good FR capabilities [1,6,23,46,48,56]. The addition of organo-modified NC improves the storage modulus of PLA in both the solid and molten stages and the polymer’s biodegradability and flexural strength [6,48,56]. NCs also can reduce the cone calorimeter test’s pHRR [6]. In a typical experiment, 3% organo-modified MMT (O-MMT) and sodium MMT (Na^+^ MMT) loaded PLA/clay NCPs were created, and their FR properties, molecular and supramolecular characteristics were investigated using a thermogravimetric analyzer (TGA), differential scanning calorimetry (DSC), and light microscopy (LM) [57]. It was determined that the filler changed the PLA matrix’s ordering at the molecular and supramolecular levels but that it encouraged char formation, decreased flammability, and increased thermal stability instead. A related study examined the melt stability and FR characteristics of O-MMT at 5 wt.% and intumescent FR (IFR) at 15 wt.% [58]. The two systems produced outstanding flame retardancy, resulting in a UL-94 V-0 rating, an LOI of 27.5%, and improved melt dripping suppression. Similar results have been published [24] for knitted fabrics finished with PLA/clay NCP in the presence of a plasticizer, in which a significant reduction in pHRR (38%) was reached. EG and O-MMT were used in synergy to create PLA-based NCPs, and their thermal and mechanical properties were assessed [33].

The synergistic effect of SiO_2_, Zinc Borate (ZnB), and O-MMT NPs in PLA aimed at retarding the flammability of PLA matrix containing 9,10-dihydro-9-oxa-10-phosphaphenanthrene-10-oxide (DOPO) has been investigated by Long et al. [59]. These researchers adopted MCD and injection molding to prepare their NCPs. The virgin PLA displayed no retardancy against fire, though upon including 10 wt.% DOPO, an improvement in the LOI from 19.1% to 27.1% was observed, along with a UL-94 rating of V-0. Additionally, with the inclusion of the O-MMT, the UL-94 rating dropped from V-0 to V-1, and the system showed melt drops during the fire test. The same was true with the NCP containing DOP, O-MMT, and ZnB. However, upon the inclusion of SiO_2_ in the composite system, the LOI value was 27% compared to the LOI value of 25.8% and 26.5% for the composite systems containing O-MMT and ZnB at 2 wt.% [59].

There is an interesting report on the utilization of C30B in synergy with EG to fabricate a PLA NCP with not just flame retardancy but enhanced thermal stability, mechanical performance and also enhanced co-dispersion of 2D NPs with PLA [33]. The improvement in the FR properties of the virgin polymer upon the inclusion of the NPs was attributed partly to accelerated PLA crystallization primarily governed by the added EG. Additionally, the enhanced thermal stability of the NCP system was attributed to the addition of organoclay. The presence of both 2D NPs was said to improve the mechanical performance and partially enhance thermal properties simultaneously. In this way, the authors prepared an NCP having either of the fillers and a ternary system having the synergized 2D NPs. Interestingly, the ternary system presented results superior to the composite systems having only clay and/or only EG, as can be observed from the UL94 results in Table 2 [33].

When C30B and EG are added to PLA, better property performance is achieved during combustion, resulting in significant reductions in sample BR compared to neat PLA. This phenomenon was postulated to be related to forming a carbonized surface layer rich in silicate or graphite, which can shield the bulk of the PLA from the heat source. These researchers also noted that PLA-EG NCPs are entirely shattered by burning and dripping, in contrast to PLA-C30B NCPs, which retain the original shape of specimens with only partial fragmentation (Figure 7a,b) [33].

Another experimental investigation conducted by Solarski and his group aimed at preparing PLA-clay NCPs for creating FR textile fibers has been explored [24]. Melt mixing was adopted in their work to process PLA and 1–10 wt.% of a particular organomodified bentonite clay (Bentone1 104-B104) in order to investigate the impact of processing variables, such as residence time, temperature, and shear on the morphology of PLA/clay NCPs. It was revealed that the dispersion of B104 occurred under various conditions without difficulty due to strong compatibility with the PLA matrix, and a comparable morphology was formed [24]. Their outcomes demonstrated that the extent of intercalation and delamination at low mixing temperatures is significantly influenced by the shear stress applied to the polymer. By adding 4 wt.% B104 to the PLA matrix during melt blending, upscale tests were carried out under improved circumstances to create NCP for the spinning process. It was then shown that, surprisingly, it is not necessary to utilize a plasticizer to spin a mixture with 4 wt.% B104 by melt spinning to obtain NCP-based multifilament yarns [24]. The yarns’ thermal, mechanical, and shrinkage properties and clay dispersion were all investigated. Even at high draw ratios, B104 may be added to PLA in amounts up to 4 wt.% without negatively affecting the tensile strength of melt-spun filaments. Interestingly, the flammability of the NCP-based multifilaments was investigated using a cone calorimeter at 35 kW/m^2^. The HRR experienced a significant decline of up to 46% [24].

Another study examined how O-MMT, an IFR, affected PLA’s melt stability and flame retardancy. A twin-screw extruder and a two-roll mill created the FR PLA. Then, adopting LOI, vertical burning test, TGA, scanning electronmicroscopy (SEM), melt flow index (MFI), and parallel plate rheological studies, the impact of IFR and MMT on flame retardancy and melt stability was carefully examined. The testing findings demonstrate that the IFR system, in conjunction with MMT, has good fire retardancy, as evidenced by the sample’s ability to receive a UL94 V-0 rating and an increase in the LOI value from 20.1 for unaltered PLA to 27.5 for the flame-retarded PLA. O-MMT greatly improves melt stability and reduces melt leaking, according to MFI and rheological measurements [58].

In another instance, clay particles were added to polylactide (PLA) together with a plasticizer, inorganic additives, plus O-MMT or unmodified O-MMT NPs (ethylene glycol) [57] to fabricate FR PLA NCP. Melt blending PLA with other components produced the PLA-based systems. While adding microparticles produced a microcomposite, combining PLA with O-MMT particles produced an NCP with an intercalated nanostructure. In both systems, the proportion of O-MMT was kept at 3 wt.%. The same blending conditions were also used to create unfilled PLA, plasticized PLA, and plasticized NCP. Poly(ethylene glycol) in the amount of 10% was utilized for plasticization. The melt filling of PLA with organomodified NPs results in an NCP with an intercalated nanostructure, as was demonstrated. In plasticized NCPs, the intercalated nanostructure was also generated. PEG molecules take part in or promote the intercalation process. Unaltered clay particles create a traditional microcomposite with a PLA matrix.

Contrary to neat PLA, it was discovered that thermo-mechanical processing of PLA improves its capacity to crystallize by heating up from the glassy amorphous form [57]. Because of the intercalated nanostructure in NCPs, including inorganic clay particles, it has a stronger inhibitory effect on PLA crystallization from the glassy amorphous state. Plasticized PLA and polymer matrices crystallize more easily in NCPs thanks to plasticizers. Independent of the sample composition, the heating technique used to induce crystallization from the glassy amorphous stage resulted in the same crystalline alteration of the PLA matrix. The sample composition impacted the spherulites’ sizes and perfection as they formed along with the crystallization. Spherulites are typically generated with smaller filler inclusions and with worse order. The plasticizer’s presence, particularly, is reported as a factor in developing delicate spherulitic morphology. In this case, the amorphous phase’s organization is said to change little with age; it is more plasticized in PLA, more constrained in NCPs, and stabilized by intercalated nanostructures [57]. It was demonstrated that intercalation is not destroyed by crystallization in NCP materials or plasticized NCP materials. Additionally, depending on the overall morphology and responsiveness to the sample mix were mechanical properties.

Among the most popular organic biodegradable polymers, PLA is utilized extensively. Its FR properties, however, are poor, as earlier stated: In order to overcome this drawback, a group of authors performed melt-blending of PLA with IFRs (melamine phosphate and pentaerythritol) in the presence of organically modified MMT (O-MMT) [6]. The authors carefully analyzed these produced nano-biocomposites with excellent intumescent char formation and improved FR properties. The MMTs utilized in this work were modified with tributyl hexadecyl phosphonium (O-MMT-2) and triphenyl benzyl phosphonium (O-MMT-1). According to a thermogravimetric investigation and Fourier Transform Infrared spectroscopy, these NCPs emit fewer hazardous gases during thermal decomposition than unmodified PLA. Concluding statements based on cone calorimeter data and char structure of different nanobiocomposites were supported by melt-rheological behaviors.

Additionally, the surfactant characteristics utilized to modify MMT were very important in regulating the fire properties of the composites. Compared to unmodified PLA, the fire characteristics of the NCP containing 5 wt.% O-MMT-1 were dramatically enhanced, with peak heat and total heat release rates (THRR) declining by 47% and 68%, respectively. In conclusion, developing high-performance PLA-based sustainable materials may benefit from the melt-blending of PLA, IFR, and O-MMT [6].

An innovative IFR PLA system has reportedly been used to study the effects of various organically modified MMTs (labeled DK1, DK2, and DK4) [4]. Lignin and microencapsulated ammonium polyphosphate made up of the IFR system were adopted, while appropriate characterization techniques were used to study the morphology of PLA/O-MMT NCPs. The composites FR and their thermal properties were assessed using the UL-94, LOI, and the cone calorimeter (CC). According to the findings, the sample containing DK2 had superior flame retardance supported by its lower pHRR and more excellent LOI value [4]. Sample PLA8 (including DK2) demonstrates a more outstanding increase in the flame retardancy of IFR-PLA when compared to DK1 and DK4, having the best LOI value of 35.3 and UL-94 V-0 at the loading of 2 wt.% of DK2. According to the cone calorimeter’s findings, the addition of IFR greatly reduces the HRR and THR of the IFR-PLA, and the addition of DK2 to IFR-PLA can further reduce the corresponding values of IFR-PLA above; however, the addition of DK2 and DK4 cannot have the same impact. According to the TGA data, the sample containing DK2, which has a higher char yield at high temperatures, has superior thermal stability than the samples containing DK1 and DK4. The TG-FTIR data demonstrate that all O-MMTs stimulated the breakdown of PLA and that the sample containing DK2 released fewer combustible gas products than the samples containing DK1 and DK4. The integration of DK2 could enhance the char quality and result in a considerably more compact and continuous morphology, as the study of the char residues demonstrates [4].

As per considered literature, it is well established that the phenomenon responsible for impacting flame retardancy to PLA-CL systems may be related to the formation of a carbonized surface layer rich in silicate, which can shield the bulk of the PLA from the heat source and the barricading of the volatiles within the composite thereby retarding degradation caused by oxidation. Additionally, considering the inexpensive nature of CLs NPs, their commercialization is achievable.

### 4.2. MXene

Through an etching process of titanium aluminum carbon (Ti_3_AlC_2_) followed by liquid exfoliation, a novel 2D nanomaterial known as titanium carbide (Ti_3_C_2_T_x_) was identified for the first time in 2011 [60,61,62]. Ti_3_C_2_T_x_ nanosheets have a wide range of potential uses in the fields of sensors, water treatment, catalysis, energy storage, electromagnetic interference shielding, and so on because of their inherently high metal conductivity, adjustable surface active sites, and notable mechanical stability [62]. Additionally, because of its low thermal conductivity and exceptional lamella thermal stability, Ti_3_C_2_Tx can have a lamella barrier effect during the pyrolysis and combustion of polymeric materials [60]. As a result, Ti_3_C_2_T_x_ has been considered a viable polymer FR additive for polymer matrices, such as PLA [60]. 

As a potential nontoxic functional ingredient for creating FR polymer composites, a novel 2D transition metal carbide called MXene (Ti_3_C_2_) has recently received much attention [36]. Mxene limits the emission of flammable volatiles while preventing further burning of the underlying polymer composite thanks to its layered flake shape, which creates a physical barrier effect. The Ti component of Mxene has been discovered to exert a catalytic attenuation effect on most polymers, unlike most traditional FRs and other 2D nanomaterials, leading to a considerable decrease in composites’ heat and smoke release [36].

Zhou et al. [36] reportedly synthesized a PLA/DOPO-Ti_3_C_2_ NCP system having excellent property performance as a FR material. The fabricated FR presented a pHRR reduction of 33.7%, a V-0 UL-94 rating owing to the DOPO-${\mathrm{Ti}}_{3}C_{2}$ ability to interplay the catalytic, barrier, and condensed effects within the PLA system. 

Cone calorimetry, the UL-94 test, and the LOI were used to demonstrate their findings on PLA composites’ thermal and burning capabilities. The authors also looked at the tensile and UV-shielding capabilities. According to the findings in the UL-94 test, PLA/${\mathrm{Ti}}_{3}C_{2}$/DOPO (3 wt.%) demonstrated a V-0 rating. The large decrease in pHRR (33.7%), total heat release (47%) and peak CO output (58.8%), as well as the improvement in fire safety, were all indicators of this (41.7%). The interaction of catalytic, barrier, and condensed effects of the Ti_3_C_2_/DOPO nanosheets in the PLA matrix, as seen in Figure 8, was the cause of the composites’ increased fire-safety performance [36]. 

In this interesting example where Mxene was adopted as reinforcing 2D NPs in the PLA system, the tensile strength of PLA/Ti_3_C_2_-DOPO increased by about 9%, and it scored “Excellent” (UPF 50+) for UV protection [36]. This study presents a novel chemical modification technique for 2D Mxene flakes to create multifunctional PLA composites, which show promise as candidates for the next wave of sustainable and protective plastic products [36].

There is an existing instance where Shi and his group [60] pointed out that Mxene (Ti_3_C_2_T_x_) can be effectively used as FR in its pristine state, and it modified form Ti_3_C_2_T_x_ (benzyldimethylhexadecylammonium chloride (HDBAC)-Ti_3_C_2_T_x_. They prepared Mxene (Ti_3_C_2_T_x_) via the etching of Ti_3_AlC_2_ using LiF/HCl at a temperature of 35 °C for 48 h, followed by washing with deionized H_2_O to neutral via centrifugation. The prepared Mxene was then dried and ready for modification with HDBAC through a facile solution mixing under a nitrogen environment for 2h, washed and dried for inclusion in the PLA matrix, as depicted in Figure 9a. 

According to Figure 9b, melt blending was used to process PLA composites. The formulations for the PLA/HDBAC-Ti_3_C_2_T_x_(HD-TC)/SiAPP composites were made by adding 0.2, 0.5, 1.0, and 2.0 wt.% of HDBAC-Ti_3_C_2_T_x_ and 15.0, 14.8, 14.5, 14.0, and 13.0 wt.% of SiAPP, respectively. Raw PLA was first vacuum-dried at 80 °C for 12 h before processing. The samples were then created using an internal mixer at 180 °C with a 50 r/min rotation speed. First, pure PLA was poured into the internal mixer. The modified Ti3C2Tx and SiAPP were introduced into the mixer for 20 min to obtain an excellent visible dispersion after melting the PLA. The sample was heated under 10 Mpa at 180 °C for additional analysis [60]. Their prepared composite/hybrid system presented the best results for their formulation: PLA/2.0HD-TC/13.0SiAPP, which contained 2 wt.% Mxene and 13 wt.% SiPPA: the LOI was enhanced from 24.4% to 33.3% with a UL-94 rating of V-0, a pHRR reduction of 49.8% along with T_5%_(°C) of 327 (residual wt.% of 10.33 at 700 °C) [60]. They investigated the residual char from the NCP in order to ascertain the FR mechanism of the PLA/Mxene/SiAPP: the formation of dense continuous intumescent C-layer was said to separate the thermal decomposition zone and hinder the transfer of heat into the underlying NCP. Additionally, the addition of HDBAC-Ti_3_C_2_T_x_ enhanced the char formation, thereby hindering the spread of the fire, as depicted in Figure 10 [60].

In their study, Huang et al. [63] prepared PLA/IFR composites with Ti_3_C_2_ Mxene nanosheets using melt blending, and they looked at the synergistic impacts of Mxene on the fire performance of PLA/IFR systems by replacing some of the IFR with Mxene. The incorporation of a small amount of Mxene might significantly improve the flame retardancy of PLA/IFR composites, according to their LOI, UL-94, and CCT results. The addition of 1.0 wt.% Mxene and 11.0 wt.% IFR resulted in a V-0 UL-94 rating, an apparent rise in LOI (160.4%), and a clear decrease in pHRR (64.6%). Intumescent char formation capability of PLA/IFR systems could be effectively improved by the presence of nano TiO_2_ catalyst and 2D nanosheet barrier effect of Mxene (as per Figure 11), thereby preventing the further flame spread and the spread of fire hazard, according to CCTs analysis, SEM, photographs, laser Raman spectroscopy (LRS), XRD, and X-ray photoelectron spectroscopy (XPS) of carbon residues of PLA/IFR/Mxene composites. It was claimed that the use of 2D Mxene in this research was successful as FR additive-based nanomaterials with self-charring and self-catalyzing functions as a high-performance synergist for various kinds of IFR polymeric matrix [63].

In summary, the 2D nanosheet barrier effect of Mxene is hypothesized to significantly increase the ability of PLA/IFR systems to create intumescent char, hence halting further flame development and the spread of fire hazards. Mxene’s layered flake form, which produces a physical barrier effect, controls the emission of combustible volatiles while inhibiting additional burning of the underlying polymer mixture.

### 4.3. Graphene-Based NPs

This sub-section presents the literature findings particularly related to graphene-based (graphite, graphene, graphene oxide (GO), and reduced GO (rGO)) FR materials utilization in PLA systems.

#### 4.3.1. Graphene

Since its separation in 2004 [39,64], graphene has attracted much interest among the 2D NMTs. A material of scientific interest in many domains, particularly in composite science, as a new FR material for different polymers, graphene offers exceptional thermal stability, gaseous and ionic impermeability, and, most crucially, non-flammable qualities [65]. GO, the precursor to graphene, comprises only one or a few layers of nanosheets, as opposed to graphite nanoplatelets, which have multi-layer graphene. This makes GO a perfect filler for enhancing the properties of polymers. Further modifications and reductions of GO with reactive oxygen functional groups, including hydroxyl, carboxyl, and epoxide groups, can be made using organic addends to improve the tortuous route effect and heat stability [65]. More importantly, low loading of modified GO will result in appreciable mechanical, electrical, and FR quality improvements because of its substantial specific surface area, superior dispersion, and exceptional mechanical and electrical properties [65]. EG, another significant layered FR material, is known to be an IFR material with graphite flakes and layers of hexagonal carbon structure that have sulfuric acid (H_2_SO_4_) intercalated between its crystalline structure [65,66]. Additionally, unlike GO, EG requires comparatively larger loading to be effective or mixed with other conventionally known FRs, which may influence the mechanical properties of PLA [1,65]. 

For instance, Murariu et al. [67] stated that PLA filled with 6 wt.% EG passed the horizontal test UL94 HB and showed non-dripping and charring development, although a 30% decrease in the pHRR was shown by cone calorimetry. 

According to Zhu and his colleagues [68], EG and ammonium polyphosphate (APP) synergistically affect flame-resistant PLA. For the PLA/APP/EG system, the ideal synergism occurs when the weight ratio of APP to EG is 1:3. In UL-94 tests, PLA composites with 15 wt.% of APP/EG (1:3) received a V-0 rating and had a 38.3% lower pHRR. These authors revealed in their study that the FR mechanism of the fabricated composite system was synergistic (Figure 12): With the passage of PLA/APP, EG initially degrades, grows, and migrates. 

The breakdown of APP follows, which generates polyphosphoric and ultraphosphoric acids and catalyzes the conversion of PLA into char residue. Inflammable gases, such as SO_2_, CO_2_, NH_3_, and H2O, are emitted during the breakdown of EG and APP in order to thin out fuels made from degradation byproducts. As a result of the flow of PLA/APP degradation products, such as polyphosphate, which promotes adhesion between graphite flakes, char layers that are continuous, dense, and sealed are produced. When heated to a high temperature, the layers retard the further deterioration of PLA, APP, and EG and slow down the volatilization of polyphosphoric and ultraphosphoric acids.

With its ability to combine the low cost and layered structure of 2D NMTs like clays with the superior thermal and electrical properties of 1D NMTs like CNTs, graphene, which can be produced in situ or easily in the form of EG nanoplatelets, may be a viable alternative to both clays and nanotubes in its use as functionalized ceramics (FRs). For several applications, especially in thermal conductors, electromagnetic interference shields, and the automotive industry, EG reinforcing filler offers outstanding competitive functional qualities to polymers like PLA (fuel injection and anticorrosion systems, fuel tank inlet, electrostatically paintable parts, etc.) [33]. 

The adopted expanded graphite, a form of graphene, as a novel 2D NMT for enhancing the non-dripping and char formation, thereby revealing an outstanding flame retardancy characteristic of PLA/EG [67]. They utilized melt processing/blending, which is an industrially scalable approach: with the inclusion of EG, the thermal stability of PLA was preserved along with enhancement in the crystallization of PLA. Furthermore, they hypothesized in their study that melt-blending PLA and EG increase the storage modulus and open the door to using these composites in applications that call for greater utilization temperatures. Cone calorimetry testing showed a drop in pHRR (approx. 30% w.r.t the virgin PLA as per Figure 13). Still, the horizontal test UL94 HB successfully passed and demonstrated neither non-dripping nor charring development [67].

It has been observed that the lack of affordable and environmentally friendly production techniques for functionalized graphene prevents its practical use in polymeric NCPs. In a particular study, for the first time, a simple but also environmentally friendly electrochemical method for producing ferric phytate functionalized graphene (f-GNS) using biobased phytic acid as both an electrolyte and a modifier was reported. Electrochemical exfoliation produces low oxidized graphene sheets with a C/O ratio of 14.8, which are tens of micrometers in size because phytic acid presence was reported. Iron and phosphorus peaks in the X-ray photoelectron spectra proved that graphene had successfully been functionalized. High-performance PLA/f-GNS NCPs can also be easily made using a practical masterbatch method. As depicted in Figure 14, a potential ferric phytate functionalized graphene FR mechanism in a PLA matrix was suggested. First, during the first deterioration, thermally stable graphene sheets function as a mass barrier to prevent the passage of combustible gases.

Meanwhile, a significant amount of phosphor carbonaceous char residue was said to be produced as a result of the phytate structure’s catalytic charring effect. In order to create an adiathermic char shield on the interior materials, high-aspect-ratio graphene was postulated to hold the char particles together. Furthermore, the suppressed reduction of gaseous products indicates that Fe-P species in the char layer can operate as a powerful catalyst for the redox reaction during the combustion process (such as hydrocarbons and CO) [69]. The high flame retardancy of PLA NCPs can be attributed mostly to the tripartite cooperative mechanism of the f-GNS [69].

#### 4.3.2. GO

Due to its outstanding thermal barrier characteristics, graphene and graphene derivatives are frequently employed as FRs in polymers. They are considered the most effective, safe materials for flame retardancy. Although chemically reduced graphene’s propensity for agglomeration and its low interfacial compatibility would degrade its mechanical capabilities, GO’s oxygen-containing functional groups can easily build linkages with their more active counterparts [70].

According to a specific study by Yu et al. [37], FR PLA composites were created by using GO as the FR and a bio-based core-shell FR (CSFR). Compared to PLA or PLA with a CSFR alone, their findings have shown that combining GO and a CSFR might give PLA more excellent thermal stability, crystallizability, flame retardancy, and mechanical qualities under the appropriate distribution. 

In this FR system, GO had an ideal loading range because of its wide range of effects, including anti-dripping, barrier, high thermal conductivity, and aggregation tendency. PLA/GO-2/4 wt.% CSFR and PLA/GO-3/4 wt.% CSFR composites had the most outstanding LOI value of 29.7% and UL 94 V-0 rating. Comparing PLA/GO-2/4 wt.% CSFR to pure PLA, the pHRR and THR values were lower by 16.7% and 6.5 wt.%, respectively [37]. GO, and the CSFR carried out their duties using the condensed-phase FR mechanism. Additionally, adding GO can make the PLA matrix more rigid and durable. PLA/GO 34 wt.% CSFR obtained improved elongation at break (7.5%), the highest notched impact strength (11.29 kJm^−2^), and improved Young’s modulus (4.1 GPa) [37].

Another study successfully engineered a dual two-dimensional (2D) graphene-derived complex to serve as an efficient, environmentally friendly FR and interface modifier for PLA materials. As a dual-purpose FR and interfacial compatibilizer, GO was first functionalized with an environmentally friendly long-chain polyester [70]. With polyester-functionalized graphene acting as a predetermined carbonization-catalyzed promoter, the 2D LDH was successfully attached. In order to give PLA matrices the desired toughness and tailored functional properties, the dual 2D LDH-anchored PG’s favorable distribution overcame the low dispersion of high-loading LDH. The findings of the experiments showed that the dual 2D graphene-derived complex might improve flame retardancy, with the HRR for the composites treated with poly(propylidene carbonate) and polybutylene succinate decreasing by 50% and 29%, respectively. The rate of CO_2_ production, total smoke production, and smoke production were dramatically decreased in PLA binary composites, which may help safeguard the environment. A new understanding of how to modify brittle polymer matrices for functional exploration was provided by the developed strategy of environmentally friendly dual 2D graphene-derived FR. It also provided a cleaner method for biodegradable composites with exceptional smoke suppression and reduced carbon dioxide emission [70].

In another study, Zhang and colleagues [71] investigated the possibility of impacting FR properties in PLA matrix by including GO, phenylphosphinic acid and nano MOFs, named zinc-modified GO (GPZ) nanohybrids. The prepared PLA included 2D hybridized NPs using a facile solution mixing/casting approach: The dried PLA was first dissolved for two hours with magnetic stirring in 40 mL of CHCl_3_. The GPZ nanohybrids were sonicated to create a homogeneous dispersion solution before being added to the PLA solution after being mixed with the appropriate amount of CHCl_3_. The mixture was agitated for four hours and then left for one. An automatic coater was used to cast the combined solution into a film. Films are roughly 0.15 to 0.05 mm thick. The PLA NCPs were dried in an oven at 50 °C for 72 h after the CHCl_3_ evaporated at room temperature to remove the remaining solvents. Their findings were enhanced thermal stability, compatibility of the GPZ with PLA matrix, enhanced mechanical properties, and outstanding improvement of the hybrid systems flame retardancy. The PLA hybrid’s flame retardancy was evidently enhanced as the pHRR of the sample (PLA-4) containing 2 wt.% GPZ showed a dramatic reduction by 39.5% (pHRR of 316.2 W/g) compared to virgin PLA with a pHRR of 523.0 W/g. The mechanism of the composites’ flame retardancy was said to be based on the formation of a graphitized char layer (via catalytic and/or cross-linking effect of GO, PPA, and zeolitic imidazolate frameworks (ZIF-8)) on which acted as a blanket during the burning as schematically shown in Figure 15 [71].

The surface-functionalized graphene NPs catalyzed the pyrolysis process of PLA, leading to the graphitization of the graphene oxide (GO) [1], which ultimately serves as a physical barrier to absorb degradation products, even though the exact mechanism for achieving flame retardancy in GO-polymer NCP systems, such as GO-PLA systems which have not been fully understood. These were catalytically transformed into a dense, cohesive char covering the polymer’s surface and preventing the transfer of heat, oxygen, and degradation products, thereby preserving the polymer.

A composite made of PLA, Cobalt(II,III) oxide (CO_3_O_4)_, and graphene has been reported, where its FR and mechanical characteristics were examined [50]. Due to graphene’s innate thermal stability, it was found that the composite PLA/CO_3_O_4_/graphene’s degradation temperature increased by 14 °C. With the addition of graphene 2D NPs, a pHRR reduction of about 40% was also achieved in the NCP, reportedly prepared by the MCD approach.

As it is well known, GO sheets (GOSs) are amphiphilic materials that have a wide range of technological applications as surfactants. Using GOSs to compatibilize immiscible polymer blends is a potential avenue because it leverages their amphiphilic property and utilizes their exceptional qualities [7]. In this regard, it has been demonstrated how GOSs’ compatibility range can be significantly increased through polymer functionalization. For instance, commercially significant blends of polyolefin-based polymeric matrices that fall outside the compatibilizing purview of unmodified GOSs may be compatible with polypropylene-grafted GOSs (PP-g-GOSs) [7], which we herewith believe this approach can be adopted for other polymers like PLA in a blend to prepare highly compatibilized polymeric composite blends or their NCPs.

#### 4.3.3. rGO

The FR mechanism of rGO is closely similar to that of GO. However, the in situ reduction of GO to rGO by heat within the PLA NCPs results in sudden expansion and increased surface area, which retards the flame or quenches it through rGO blanketing of the flame. When only GO and/or organic hybrid FRs are used, they exhibit a synergistic FR action between themselves. Thus, the physical barrier function of GO in hybrid FRs is crucial in preventing the thermal degradation of polymers such as PLA, even as organic RF on the surface of GO thermally decomposes into char during the initial combustion stage, gradually developing into a continuous and compact char layer with GO, thereby reinforcing the barrier effect of GO.

For instance, ${\mathrm{CO}}_{3}O_{4}$/graphene composites with tricobalt tetraoxide functionalization were created to lessen the fire risks associated with aliphatic polyesters. The ${\mathrm{CO}}_{3}O_{4}$/graphene was characterized using X-ray diffraction (XRD), Raman spectroscopy, transmission electron microscopy (TEM), and atomic force microscopy (AFM), which supported the chemical structure. Both poly(butylene succinate) (PBS) and PLA incorporated ${\mathrm{CO}}_{3}O_{4}$/graphene, which increased the initial degradation temperature and retarded the thermal decomposition process. Comparing the HRRs of PBS-${\mathrm{CO}}_{3}O_{4}$/graphene and PLA-${\mathrm{CO}}_{3}O_{4}$/graphene composites to pure PBS and PLA were lowered by 31% and 40%, respectively. Additionally, the presence of ${\mathrm{CO}}_{3}O_{4}$/graphene considerably reduced gaseous products, such as hydrocarbons, carbonyl compounds, and carbon monoxide. This is because of the material’s strong catalytic activity for the oxidation of CO in combination with its barrier effect qualities [50].

A compatibilizer functionalized with pyrene was created by one-step ROP to manufacture graphene/PLA composites with good performance. This compatibilizer demonstrated significant π−π interaction with graphene. Using a solution-casting technique, composite materials containing uniformly scattered graphene could be produced due to the dispersion stability of graphene in organic solvents. The compatibilizer positively affected the composites’ mechanical, thermal, and crystallization properties for the resulting graphene and PLA composites. Additionally, the use of graphene significantly boosted the electrical conductivity of the polymer matrix, while the compatibilizer did not significantly alter the electrical conductivity of the resulting composites. The findings suggested that the compatibilizer has significant potential for use in polymer and carbon allotropes composites. It was generated using the straightforward synthetic method described here [5].

The application of azo-boron (AZOB) modified rGO intercalated by sodium metaborate (SMB) for simultaneous suppression of smoke and hazardous fumes, increased tensile strength, and FR of PLA NCPs has been reportedly studied in a paper. By aryl grafting AZOB onto GO and then in situ reducing/intercalating it with sodium borohydride/SMB, the rGO-AZOB/SMB hybrid was created. After that, the rGO-AZOB/SMB hybrid characteristics in the PLA matrix were studied. Cone calorimeter testing shows enhanced FR performance with notable drops in pHRR (76.5%), total heat release (76.9%), total smoke release (55.6%), peak CO production (25.9%), and peak CO_2_ production (78.6%). With a higher LOI value of 31.2%, a V-0 rating was obtained in the UL 94 test. There were notable drops in pyrolysis products, particularly hydrocarbons, CO, CO2, and carbonyl compounds. Tensile strength and Young’s modulus increased by 34.9% and 49.1%, respectively. According to previous research, the glassy charring effect of AZOB’s B-OH groups and rGO’s intrinsic lamellar blocking effect serve as the main foundations for the FR mechanism of rGO-AZOB/SMB/PLA NCPs (Figure 16) [65].

It is well known that the aspect ratio of the utilized 2D NPs used in any composite system greatly influences PLA/its blend matrix property performance. Thus, for easy correlation of the surveyed 2D NPs, we also present in Table 3 the aspect ratio of the diverse 2D FRs with the flame retardancy of their host matrix (PLA and/or its blends). Though most of the literature failed to mention the aspect ratio of the 2D NPs utilized in their studies, this proves that there is a daring need to investigate the influence of 2D NPs aspect ratio on its FR properties in future research.

From Table 3, we have observed that the UL 94 rating for graphene and its derivatives is thus: V-0 and, in some instances, V-2, but also HB; clay NPs: V-0 on the average and in limited instances V-2, limit as well as HB; MXenes: generally V-0 has been reported; LDH: V-0 only; and MoS_2_: Reports on UL 94 are not yet available, but these 2D NP have shown good thermal stability and a decrease in carbon dioxide/monoxide (CO_2_/CO) emission during pyrolysis of over 90% when compared to pure PLA. It can be deduced that the FR properties of PLA-2D NPs NCPs are outstanding, but there is still room for improvement. 

On the other hand, the LOI of the reviewed PLA-2D NPs fabricated NCP systems as per Table 3 have also been interesting, with graphene and its derivatives usage presenting LOI between 24% and 36.5%, clay systems LOI is observed between 23% and 42%, MXenes systems LOI observed between 32.7% and 33.7%, and LDH based system present LOI value between 29% and 38.9%. These systems do not support burning; their LOI values are far beyond 21%. Therefore, researchers can carefully choose 2D FR NPs for their predetermined application with reference to Table 3.

### 4.4. LDH

LDHs are lamellar inorganic solids that resemble hydrotalcite and have a brucite-like structure. Trivalent cations partially substituted for divalent cations result in a positive sheet charge counterbalanced by anions in the interlayer galleries. Among 2D materials, LDHs are one type. LDHs are classified as anionic clay and consist of cationic layers that resemble brucite, with trivalent cations introducing an overall positive charge to the nanosheets. The interlayer galleries leading to the generalized LDH formula contain charge compensating anions; [M^2+^_1−*x*_M^3+^*_x_*(OH)_2_]*^x^*^+^[A*^n^*^−^]*_x_*_/*n*_·mH_2_O, where M^2+^/M^3+^ = divalent/trivalent metal cation and A*^n^* = interlayer.

While LDHs and layered silicates like MMT have layered crystalline structures with substitutable ions in the interlamellar region, their chemical and structural characteristics, such as composition, geometries, and layer thickness, are different. LDHs, as previously indicated, are anionic clays because anions are present in the interlamellar gallery of positively charged layers. In contrast, layered silicates exhibit a reversed structure, hence the term cationic clays.

In contrast to layered silicates, which have two or more metal oxide sheets in a sandwiched structure, LDHs only have one octahedral metal hydroxide sheet, making up each crystal layer. A single crystal layer in MMT comprises two silica tetrahedral sheets and an octahedral sheet comprising Fe, Al, and Mg. As a result, LDHs have less stiffness and thinner crystal layers than layered silicates [85]. Figure 17 provides a schematic representation illustrating the structural and chemical distinctions between LDH and MMT [85].

The layers of LDHs primarily contain divalent and trivalent metal cations from the third and fourth periods of the periodic table. The typical trivalent ions present in DLH layers are Ca^2+^, Ni^2+^, Mg^2+^, Co^2+^, Zn^2+^, Mn^2+^, Fe^2+^, Ti^2+^, Cu^2+^, Cd^2+^, etc., while the trivalent metallic ions typically found in LDH layers are Ga^3+^, Al^3+^, Fe^3+^, Cr^3+^, Mn^3+^, V^3+^, Y^3+^, In^3+^, Ru^3+^, La^3+^, etc. Halides-based anions, such as chloride, fluoride, and so on; oxoanions like carbonates, bromates, nitrates, sulfates, etc.; anionic complexes like ferricynide and ferrocyanides, (PbCl_4_)^2−^, polyoxo- and oxo-metallate, such as chromate, dichromate (Mo_7_O_24_)^6−^, (V_10_O_28_)^6−^ etc., as well as alkyl sulfates, phosphonates, and carboxylate based upon organic anions, etc. are typically present as interlamellar anions in LDHs.

Due to their distinct molecular conformations and layered structure, LDHs are a new class of FR chemicals with promising futures in fire chemistry and smoke reduction [1]. The loss of interlayer water, intercalated anions, and dihydroxylation to generate metal oxides, which absorb a lot of heat and lower oxygen concentration, are all part of the FR and smoke suppression mechanism of LDHs [1]. This process encourages the growth of carbonaceous char on the polymer, shielding the bulk of the polymer from the air and reducing smoke production from asphyxia. It has been shown that the presence of 2% Zn-Al-LDH in the PLA/FR matrix decreased pHRR by 58% (Figure 18) in both the microscale combustion calorimeter (MCC) and the cone calorimeter. The PLA NCPs were made from organo-modified zinc-aluminum (Zn-Al-LDH), ammonium polyphosphate, melamine cyanurate, and pentaerythritol [52].

In another investigation, Shan et al. processed a novel PLA-based FR NCP using HPCP/LDH-SDS in the PLA [82]. According to these scientists, PLA/HPCP/LDH-SDS composites’ thermal analysis and char residue study revealed that different LDH-SDS materials, including NiFe, NiAl, and NiCr, play diverse roles in enhancing the thermal stability and flame retardancy of PLA composites. The effective mechanism of LDH-SDS and HPCP in PLA composites was also hypothesized to work as follows: Layers of LDH-SDS act as a physical barrier that can slow down heat and mass transfer between the gas and the condensed phases, shielding the underlying material from further burning and lowering its heat release. Acid species or metal oxides can catalyze esterification reactions between PLA and polyol phosphate compounds. This helps the matrixes’ thermal stability by enhancing it [82].

On a similar note, another group, [30], improved upon the work reported by Shan et al. [82]: Shan and his colleagues, in their work, revealed that though they could enhance the FR properties of their formulated systems, the mechanical properties were diminished significantly. In order to address the obvious mechanical flaws associated with FRs on PLA that contain LDH, IFR composed of silane-coated ammonium polyphosphate (APP), pentaerythritol phosphate (PEPA), and nickel aluminum layered double hydroxide (NiAl LDH) with cornstarch (CS), and PBS as mechanical reinforcers, was investigated [30]. Excellent FR characteristics were present in the PLA/PBS/APP/CS/NiAl LDH composites. The PLA/PBS/APP/CS/NiAl LDH composites’ mechanical properties (i.e., tensile strength and elongation at break) improved with more PBS and CS content in the matrix. To avoid compromising the mechanical qualities for FR gain, it is advised that the LDH content in the PLA polymer matrix be decreased.

The utilization of PA-modified LDH, along with the addition of diethylenetriamine penta-(methylenephosphonic) (DT-M: prepared via self-assembly) acid as an FR agent in PLA has been exploited by Jin and colleagues [83]. These authors first prepared a modified form of LDH by intercalating PA within the pristine LDH layers before the NCP preparation. Upon inclusion of 1% PA@LDH and 14% DT-M in virgin PLA in the fabricated NCP, they observed a dramatic improvement of 19.4% to 38.9%, V-0, and 63% (i.e., 812 to 301 kW/m^2^ as in Figure 19 below) for LOI, UL-94 rating, and % reduction in pHRR, respectively. The residual char was also observed to increase from 0.8 to 10.6% at 700 °C.

The FR mechanism proposed by these authors, as presented in Figure 20, was described thus: DT-M quickly breaks down into DTPMPA and MEL when the PLA composites are heated. Since MEL serves as the primary gas source in the IFR system, ammonia may be quickly released from it in the gas phase. Moisture can be released from either the interlayer water of PA-LDH or the cross-linking of polyphosphoric acids. The cross-linking of broken-down DTPMPA is the main source of the char residues in the condensed phase. Additionally, the char formation process involves the breakdown products of MEL (mostly melem and melon) and PA-LDH, particularly for PA-LDH. This enhanced LDH, which has wider interlayer spacing and improved dispersion, offers more polyphosphoric acid reaction sites, creating more char residue and improving the fire resistance of PLA composites. In addition, the breakdown byproducts of the DT-M and LDH layers can also be reacted with by the intercalated PA anions. The compact char expanded due to the discharge of combustible gases in combination with it. As a result, the char formation process was primarily influenced by the DT-M through the release of nonflammable gases and the promotion of char formation. At the same time, the PA-LDH could contribute by chemically interacting with DT-M degradation products, as per Figure 20.

The solution exfoliation and film casting methods were used in one study to create PLA/LDH films with good FR properties and transparency. By carefully designing the inorganic-organic interfacial interaction between NiAl-LDH and PLA, 2-carboxylethyl-phenyl-phosphinic acid (CEPPA) was able to effectively address the dispersion issue of NiAl-LDH nanolayers and obtain improved PLA flame retardancy in the composites. According to their findings, the PLA matrixes’ NiAl-LDH/CEPPA (LC) nanolayers exhibited exfoliated structures and were evenly distributed. Even with LC content up to 10% by weight, all PLA/LDH films had excellent transparency. UV light was absorbed by the PLA/LDH films, which lessened the embrittlement of PLA films. From 12.0 kJ/g of virgin PLA, the sample with 10 wt.% LC’s total heat release (THR) dropped to 9.7 kJ/g [84].

With regards to the adoption of LDH and its compounds as FRs, scientists have found that when LDH is utilized as an FR filler, polymer/LDH NCPs will be implanted into increased barrier characteristics by obstructing diffusion of volatile decomposition products, carbon monoxide, and smoke yields. Dehydration of LDH and encouraged char formation during later heat treatment may also improve the materials’ fire safety at the same time. 

### 4.5. Others

Reports on the usage of FRs 2D layered NPs other than those earlier discussed in PLA systems are very rare or highly limited. However, research in novel 2D materials like graphdiynes, Xenes (single-element graphene analogs), MoS_2_, etc. application as FRs in PLA is believed to be underway by several groups, though challenged by the cost of these new 2D NPs/NMTs.

Few-layered MoS_2_ and MoS_2_/M_x_O_y_/CNT nanomaterials have been synthesized and added to the PLA composites using twin-screw extrusion mixing in an interesting study to improve the fire performance of PLA. They fabricated samples containing 0.5 wt.% of few-layered MoS_2_ and 2 weight percent of MoS_2_/Co_2_O_3_/CNT; TGA analysis demonstrated a 40 °C and 65 °C drop of T_max_ value. The creation of a charred residue layer caused by the addition of a few-layered MoS_2_ allowed for a reduction in CO emission during the burning of up to almost 70% in the case of a sample containing 1 wt.% of MoS_2_. By adding MoS 2/Fe2O3/CNT to the PLA matrix, these groups accomplished a reduction in CO emission of over 90% [34]. Few-layered MoS_2_ and MoS_2_/Fe_2_O_3_/CNT composites performed best, according to MCC analysis. In contrast, MoS_2_/Co_2_O_3_ and MoS_2_/Ni_2_O_3_/CNT composites showed strong effects of the secondary stage that occurs before the main PLA combustion, as well as barrier instability at 0.5 wt.% and 1 wt.% of FR. Due to the presence of CNT, all of the composites had improved thermal conductivity when compared to pure PLA, up to a maximum of 65% in the case of a composite comprising an addition of 2 wt.% of MoS_2_/Fe_2_O_3_/CNT. In summary, the introduction of few-layered MoS_2_ and CNT-functionalized MoS_2_ NMTs demonstrates good potential for lowering the flammability of the PLA, which may be advantageous for both academic research and real-world applications [34].

It was clear that the inclusion of MoS_2_ in the PLA matrix systems retards its flammability by creating a barrier effect that effectively prevents oxygen filtration, heat, as well as mass transfer, and the release of combustible gases created during the combustion process.

## 5. Applications

The application of 2D layered NPs converses in this article has been broad and is presented herewith. As per the literature, researchers have suggested the following area as the potential niches of PLA included 2D NPs NCPs. Table 4 shows some of the potential/suggested niches of 2D NPs NP-reinforced PLA NCPs.

It is impossible to exhaustively discuss the application areas of the discussed 2D NPs NP-reinforced PLA. Still, it is known that For several applications, especially in thermal conductors, electromagnetic interference shields, and the automotive industry, EG reinforcing filler offers outstanding competitive functional qualities to polymers like PLA (fuel injection and anticorrosion systems, fuel tank inlet, electrostatically paintable parts, etc.) [33]. 

## 6. Challenges

Even though we have seen, thus far, that the utilization of 2D NPs in PLA systems presents excellent FR properties, some challenges still exist, such as reduced mechanical properties of the concerned NCP systems in most cases. Researchers have postulated that this challenge could be overcome by adding nanofibres as reinforcing filler materials. Another way to overcome this challenge may be by blending PLA with other thermoplastic polymers with better mechanical properties.

In some literature, another concern is the use of toxic FR compounds, such as halogenated FRs, in conjunction with these 2D NPs in PLA systems. However, this issue can be solved by utilizing more eco-friendly FR additives like intumescent FRs in synergy with the 2D NP adopted as the active filler in the PLA matrix.

It is vital to note that reports on the usage of FRs 2D layered NPs other than those discussed in PLA systems are rare or highly limited. However, research in novel 2D materials like graphdiynes, Xenes (single-element graphene analogs), MoS_2_, etc. application as FRs in PLA is believed to be underway by several groups, though challenged by the cost of these new 2D NPs/NMTs. The challenge of the high cost of novel 2D layered NMTs greatly limits their application in biopolymeric NCP systems aimed at manufacturing commodity products consumed by middle- and lower-class consumers. Researchers are encouraged to develop novel and facile approaches for industrial-scale synthesis of these 2D NPs. 

Other challenges researchers face requiring immediate solutions are the difficulties encountered with robust functionalizing target surface functionalities on the surface of 2D NPs, especially graphene-based.

Also, most synthesis/modification approaches for 2D, such as MXenes, LDH, or graphene-based NPs, are complex and expensive. Researchers in these niches should focus on developing facile synthesis/modification approaches that can be scaled for commercial production.

The recovery and/or conversion of synthesis by-products obtained from the synthesis of these 2D NPs is rare/limited or not properly discussed in currently available literature except for NCs and partially for graphene and its derivatives. We recommend that researchers discuss these issues raised herewith properly in their works.

We have observed that in some works, important analysis like UL 94, LOI, smoke suppression, and/or TGA is not considered: it is expedient that researchers holistically analyze PLA NCPs filled with these active 2D NPs aimed at applications where flame retardancy is key.

It is important to remember that FRs like the ammonium polyphosphate (APP) based intumescent system have been widely employed and shown to be quite efficient at making PLA FR, they are not without their drawbacks [49]. The U.S. Environmental Protection Agency (EPA) has determined that APP has a very high grade for environmental persistence. The fillers used in PLA composite materials must be safe for the environment because they will be disposed of with the PLA in landfills. Consequently, employing APP in the PLA system is incompatible with our “environmentally sustainable” goal. There have been suggestions for melamine polyphosphate (MPP), which, according to the EPA, is less persistent and does not bioaccumulate in the environment. The FR polymer industry is increasingly interested in MPP, a halogen-free, nitrogen-containing, phosphorus-based FR agent. During burning, the melamine component of MPP sublimes by absorbing a considerable amount of heat and emitting a nitrogen-rich gas to saturate the surrounding air with nitrogen. In contrast, the polyphosphate component promotes char formation by promoting the polymer chain dehydration [49]. The enhancement of the flame retardancy, as well as the mechanical and other functional properties of PLA NCPs for use in advanced materials, is hypothesized to result from the synergistic combination of MPP and 2D layered NPs or the attachment of MPP onto 2D layered NPs towards its inclusion as reinforcing filler in PLA systems.

Last but not least, most literature purposefully omitted or neglected to specify the aspect ratio of the 2D NPs used in their investigations, demonstrating the urgent need for further research into the impact of the 2D NP aspect ratio on its FR properties.

## 7. Conclusions and Prospect

### 7.1. Conclusions

The twenty-first century has shown the value of nanotechnology, which has already produced several materials for use by humans. The best examples of the miracles that nanotechnology has brought about, and there are yet more to come, include 2D NMTs like NCs, graphene, MXenes, and others. Materials with two dimensions that are not nanoscale are known as 2D NMTs. These nanomaterials feature strong intralayer covalent bonds and weak interlayer van der Waals bonds, which give them exceptional electrical, optical, mechanical, and other properties. They also have thin layers and single-layer crystals. With remarkable physical, chemical, and mechanical characteristics, this class of NMTs has a form like a plate. In contrast to conventional reinforcing fillers like carbon black, 2D NMTs have become very popular due to their exceptional properties. Despite this, they are already finding use in various advanced materials applications, including tissue regeneration, topological insulators, sensors, batteries, photodetectors, transistors, EMI shields, sensors, and FRs.

According to the literature under consideration, it is well known that the phenomenon affecting the flame retardancy of PLA-CL systems, for example, may be connected to the formation of a carbonized surface layer rich in silicate, which can shield the bulk of the PLA from the heat source and the barricading of the volatiles within the composite, thereby delaying degradation brought on by oxidation. Additionally, given the low cost of CLs NPs, their commercialization is feasible. The graphene-based 2D NPs RFs technique in PLA is mentioned as having a similar mechanism. However, the challenge encountered with nanoclay is that it requires chemical modification in instances where the formulated polymeric NCP is made from a blend of immiscible polymers (PLA/other polymers) and/or in a hybrid system where more than one active 2D NPs is adopted.

According to the literature review, the 2D nanosheet barrier effect of MXene is hypothesized to dramatically boost the ability of PLA/IFR systems to produce intumescent char, hence preventing further flame growth and the spread of fire hazards. The physical barrier effect that the layered flake form of MXene creates regulates the emission of flammable volatiles while preventing further burning of the underlying polymer mixture.

Scientists have discovered that when LDH is used as an FR filler, polymer/LDH, NCPs will be implanted into increased barrier characteristics by obstructing diffusion of volatile decomposition products, carbon monoxide, and smoke yields. This is relevant to the adoption of LDH and its compounds as FRs. Dehydration of LDH and enhanced char formation after subsequent heat treatment may also simultaneously increase the fire safety of the materials.

Again, it was obvious that adding MoS_2_ to PLA matrix systems reduces flammability by acting as a barrier to oxygen filtration, heat transfer, mass transfer, and the release of flammable gases produced during combustion.

The preparation of these 2D NPs reinforced PLA FR NCPs has been largely by MCD/intercalation, with few reports on solution mixing and film cast, melt spinning, electrospinning, and, lately, 3D printing. The adoption of MCD is largely due to the ease of processing 2D NPs with PLA at an industrial scale. However, developing novel approaches where even unmodified 2D NPs could be effectively utilized is greatly required.

This article presents an exclusive, though not exhaustive, report on the progress in utilizing the most important 2D NMTs as FRs in PLA-based NCP systems, focusing on their processing, applications, challenges, and future perspectives.

### 7.2. Future Prospects

The future of 2D NPs application as FR in PLA NCP systems and other biopolymeric or non-biopolymeric matrices is bright. Researchers have presented excellent findings even as presented in this literature. However, there is a need for more exploitation of these systems, with PLA seeing the world is headed toward greener materials as alternatives to their synthetic counterparts.

For instance, there are just a few studies examining the effect of NP shape (spherical, fibrillar or tubular, plate-like) on the fire resistance of polymer NCPs where CNTs and layered silicate were used in nylon 6 matrix [86], sepiolite NCs and MWCNTs were adopted in synergy in PLA system [78], and the quantity is even more restricted for PLA functionalized/reinforced 2D NPs [53,78]. A potential synergistic combination of FR chemicals and nanofillers has hardly ever considered the influence of the nanofiller geometry [53]. A NASA-formulated IFR ingredient in PS has reportedly been employed in conjunction with spherical (fume silica) and rod-like (exfoliated attapulgite) nanofillers in one specific case [16]. While both fillers reduced the HRR, the LOI performance was affected differently. The scientists noted that combining spherical and rod-like nanofillers is an efficient way to increase fire retardancy. As researchers working on the design and fabrication of PLA/2D NP composite systems, we need to consider investigating the influence of NP geometry and the effect of amalgamating these geometries on the effective FR properties of these composite systems.

More research should be performed to clarify the destiny of FRs in disposal/recycling activities. Further analyses of FRs are also required. The volume of usage, inherent harmful effects on human health and the environment, exposure assessments, persistence, and bioaccumulation/biomagnification of FRs or their breakdown products are all useful factors for determining priorities. An international conference might consider standardized labeling, given the rise in the recycling of FR products. It is best to avoid using FRs that are persistent and bioaccumulative.

Nothing is known about how quickly FR nanofillers biodegrade, particularly with additional FR compounds, to create more potent effects. As a result, we still have a challenge with using 2D NPs in PLA systems that are completely ecologically friendly. These findings open up the possibility of expanding the use of various plant cultures, particularly energy crop cultures, as valuable, sustainable sources for PLA-2D NPs NCPs within biopolymeric composites research projects while at the same time creating biodegradable FR 2D NPs to add flame retardancy to advanced biopolymeric matrices like PLA, polyhydroxybutyrate (PHB), bio-PET, etc.

The future is promising for applying advanced functional materials with excellent flame retardancy with the utilization of novel 2D NPs and their emerging counterparts in PLA-based NCP systems.

## Figures and Tables

**Figure 1 materials-16-06046-f001:**
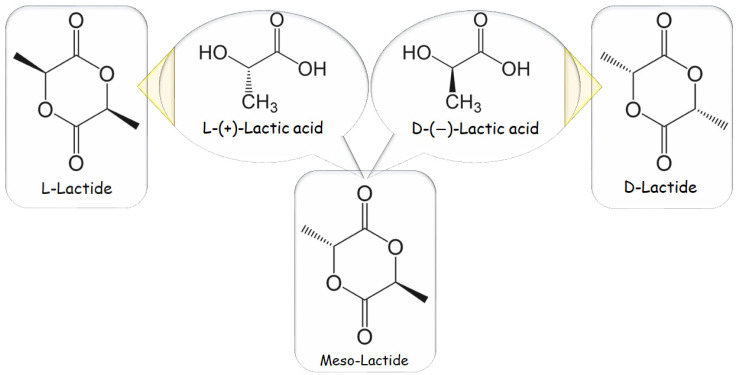
Lactic acid and lactide are initial stereoisomers.

**Figure 2 materials-16-06046-f002:**
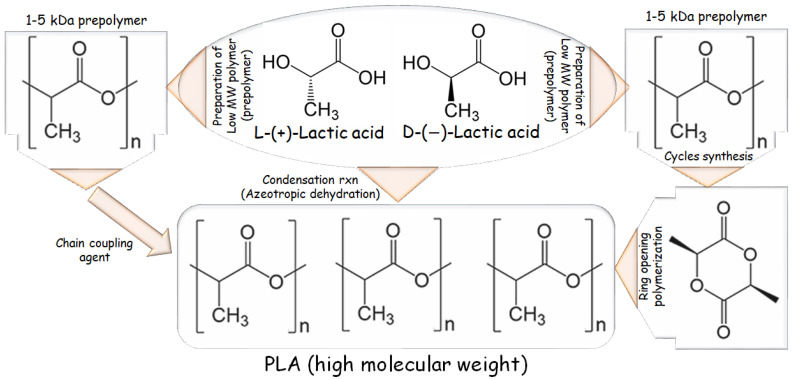
Major production routes for PLA.

**Figure 3 materials-16-06046-f003:**
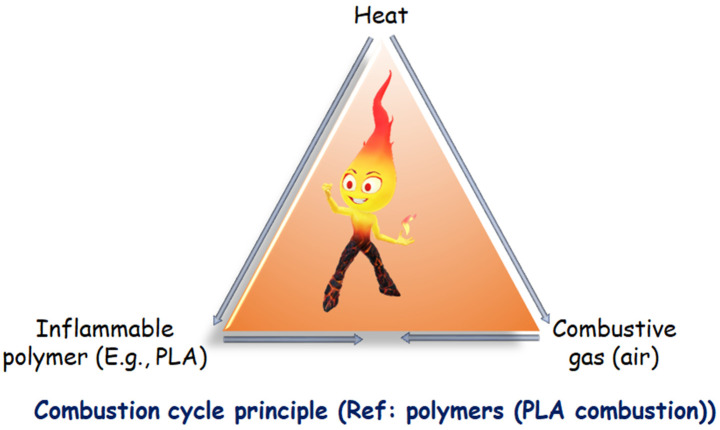
Combustion cycle principle with reference to PLA.

**Figure 4 materials-16-06046-f004:**
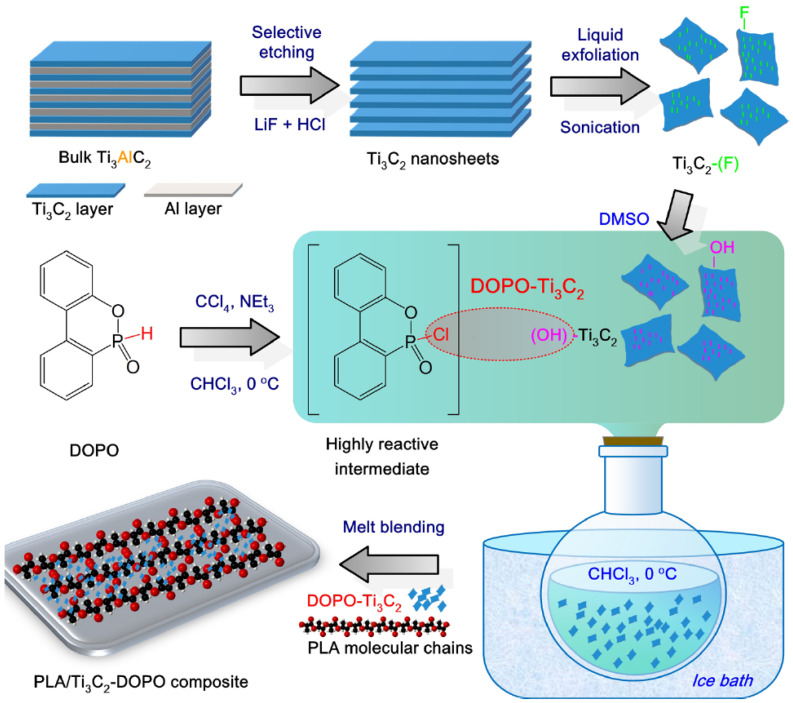
Synthesis of Ti_3_C_2_–DOPO for multifunctional PLA composites. Reproduced with permission from Zhou et al. [36], Copyright 2021, ACS Publications.

**Figure 5 materials-16-06046-f005:**
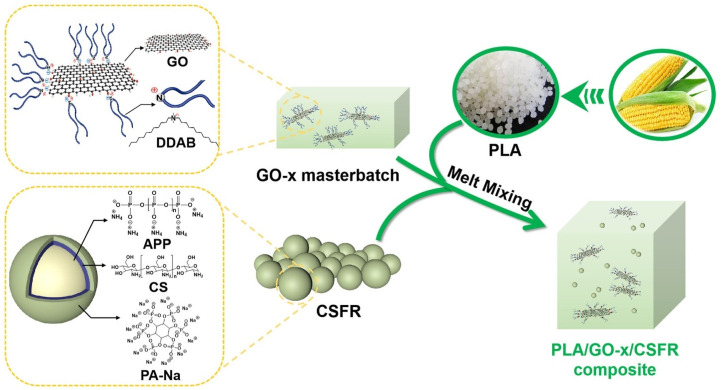
The preparation of PLA/GO-x/CSFR composites. CSFR, core-shell FR; PLA, polylactide; GO, graphene oxide. Reproduced with permission from Yu et al. [37], Copyright 2023, Elsevier Science Ltd.

**Figure 6 materials-16-06046-f006:**
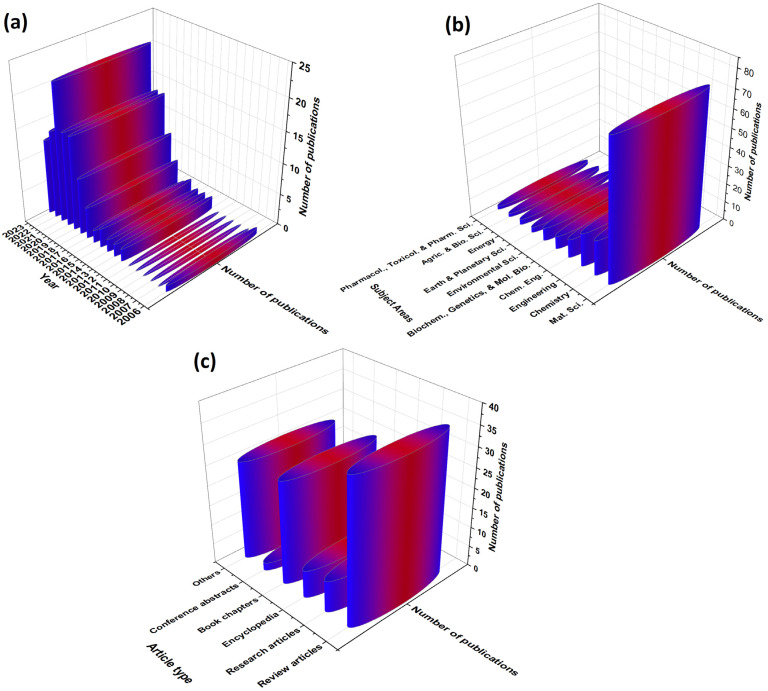
Progress in 2D NPs-PLA RF NCP systems for diverse niches. (**a**) number of publications per year, (**b**) Number of publications with respect to article types, and (**c**) Number of publications with respect to subject area. With a search on Science Direct on 29 June 2023, using the keywords “2D nanomaterials PLA FR materials”.

**Figure 7 materials-16-06046-f007:**
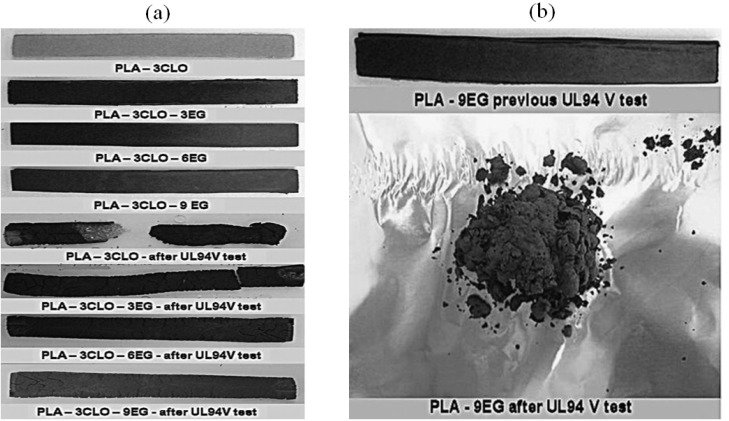
(**a**) Images of neat PLA and PLA specimens containing CLO, previous and after UL94V test and (**b**) Images of PLA-9EG specimen before UL94V testing and residue obtained by burning. Similar observations were obtained for PLA-3EG and PLA-6EG NCP. Reproduced with permission from Fukushima et al. [33], Copyright 2010, Elsevier Science Ltd.

**Figure 8 materials-16-06046-f008:**
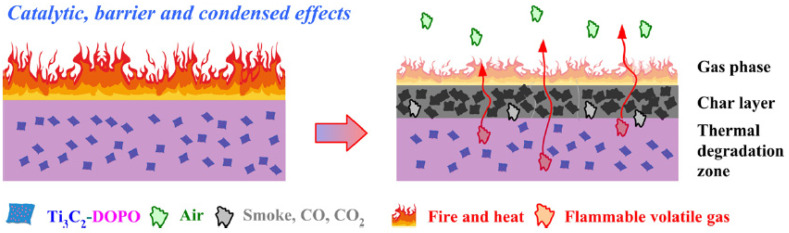
Proposed mechanism of FR. Reproduced with permission from Zhou et al. [36], Copyright 2021, ACS Publications.

**Figure 9 materials-16-06046-f009:**
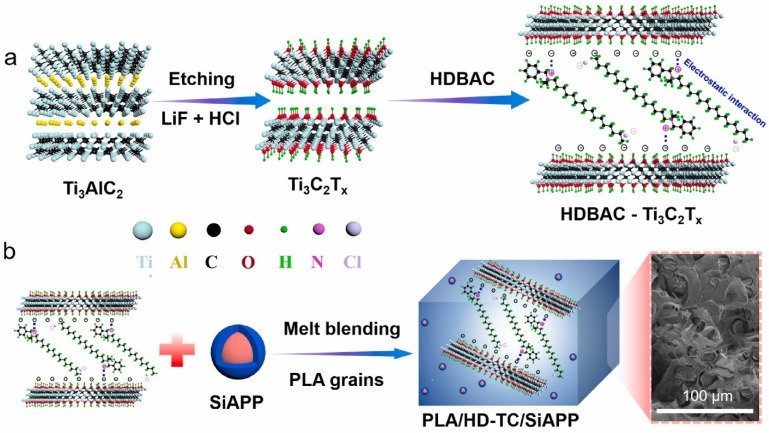
Schematic diagrams for HDBAC-Ti_3_C_2_T_x_ (**a**) and PLA/HD-TC/SiAPP composites (**b**). Reproduced with permission from Shi et al. [60], Copyright 2022, Elsevier Science Ltd.

**Figure 10 materials-16-06046-f010:**
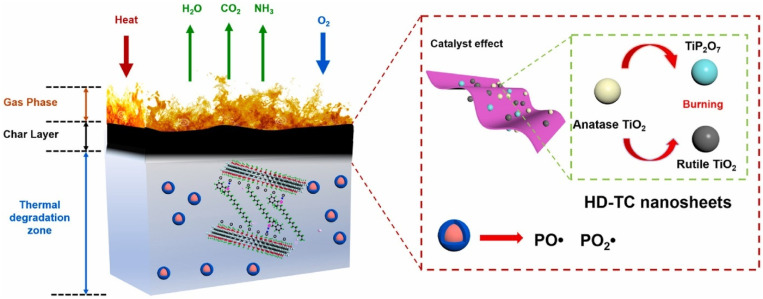
Schematic illustration for the proposed flame-retardant mechanism. Reproduced with permission from Shi et al. [60], Copyright 2022, Elsevier Science Ltd.

**Figure 11 materials-16-06046-f011:**
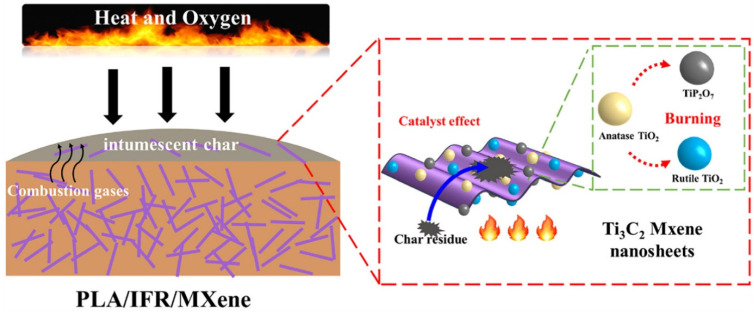
Schematic illustration for the synergistic flame retarded mechanism of PLA/IFR/Mxene NCPs. Reproduced with permission from Huang et al. [63], Copyright 2020, Elsevier Science Ltd.

**Figure 12 materials-16-06046-f012:**
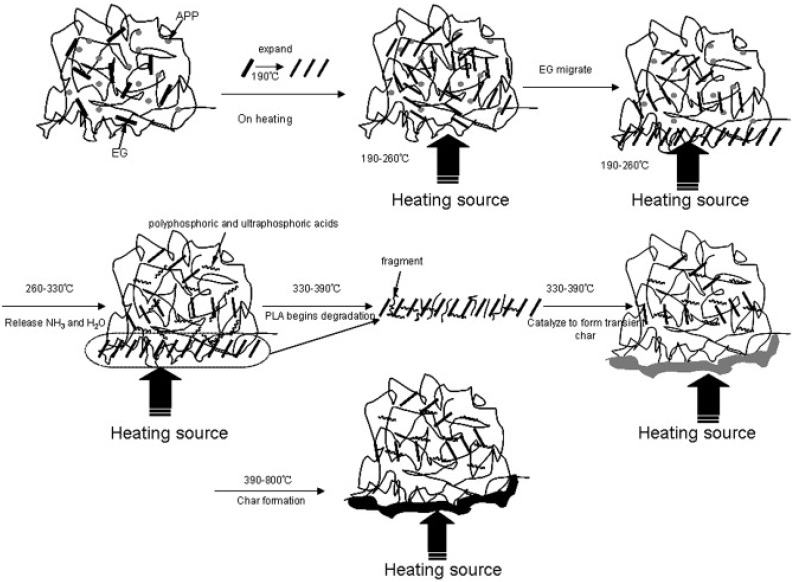
FR mechanism of PLA/APP/EG composite. Reproduced with permission from Zhu et al. [68], Copyright 2011, Elsevier Science Ltd.

**Figure 13 materials-16-06046-f013:**
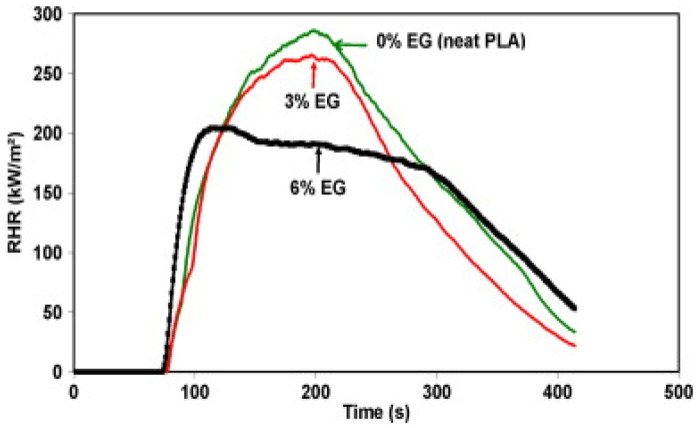
RHR plotted against time: neat PLA compared to PLA–EG composites with different amounts of EG nanofiller. (Color on the Web). Reproduced with permission from Murariu et al. [67], Copyright 2010, Elsevier Science Ltd.

**Figure 14 materials-16-06046-f014:**
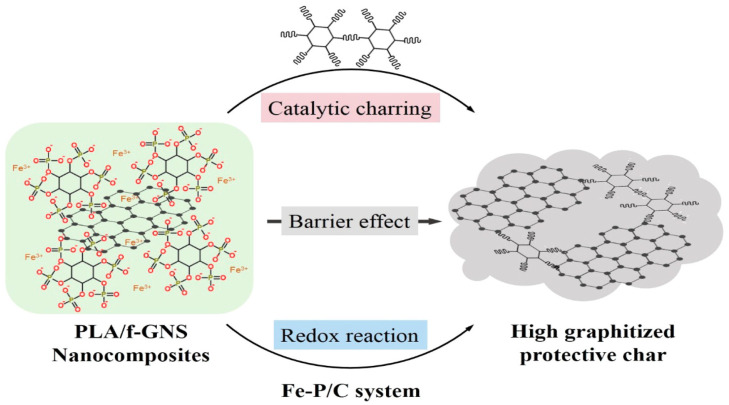
Scheme of proposed flame-retardant mechanism for f-GNS in PLA NCPs. Reproduced with copyright permission from Feng et al. [69], ACS Publications.

**Figure 15 materials-16-06046-f015:**
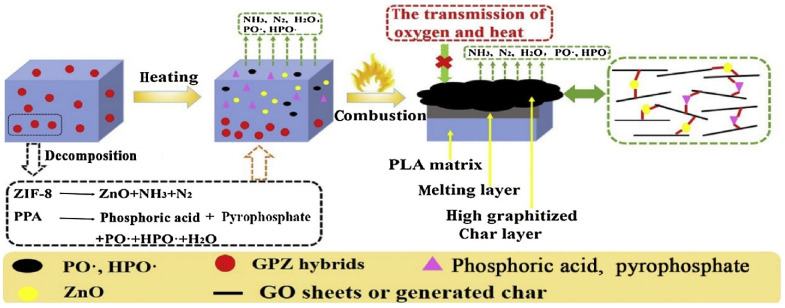
Possible FR mechanism of PLA NCPs. Reproduced with permission from Zhang et al. [71], Copyright 2020, Elsevier Science Ltd.

**Figure 16 materials-16-06046-f016:**
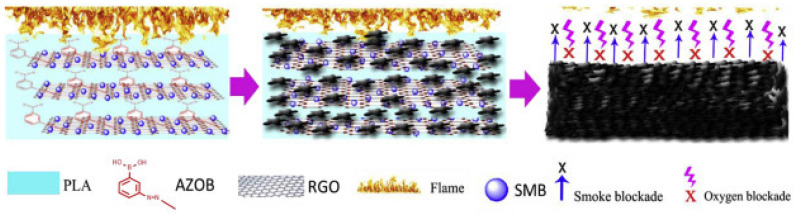
Proposed FR mechanism of RGO-AZOB/SMB/PLA NCPs. Reproduced with permission from Tawiah et al. [65], Copyright 2019, Elsevier Science Ltd.

**Figure 17 materials-16-06046-f017:**
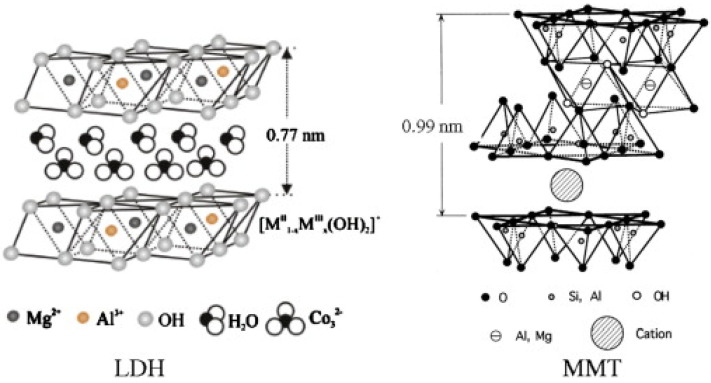
Schematic diagram showing the structural and chemical difference between LDH and MMT. Reproduced with permission from Das et al. [85], Copyright 2008, Elsevier Science Ltd.

**Figure 18 materials-16-06046-f018:**
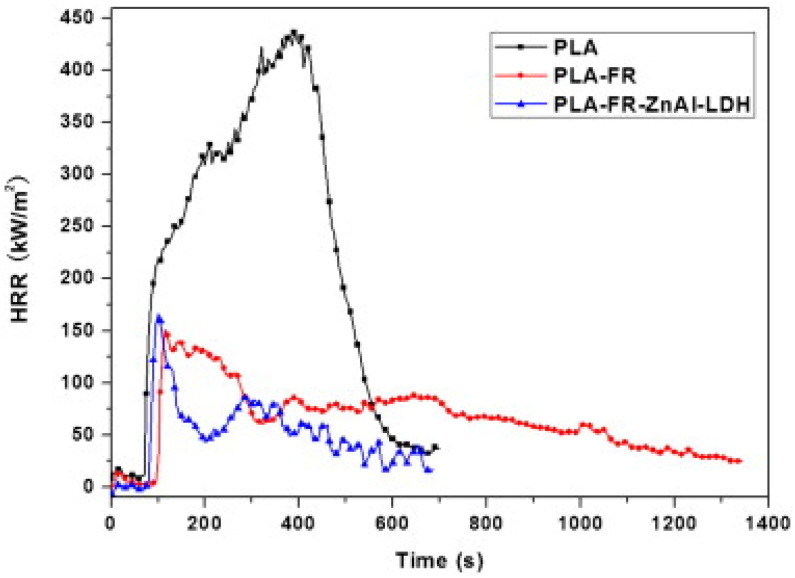
Heat Release Rate (HRR) for samples from cone calorimeter test. Reproduced with permission from Wang et al. [52], Copyright 2010, Elsevier Science Ltd.

**Figure 19 materials-16-06046-f019:**
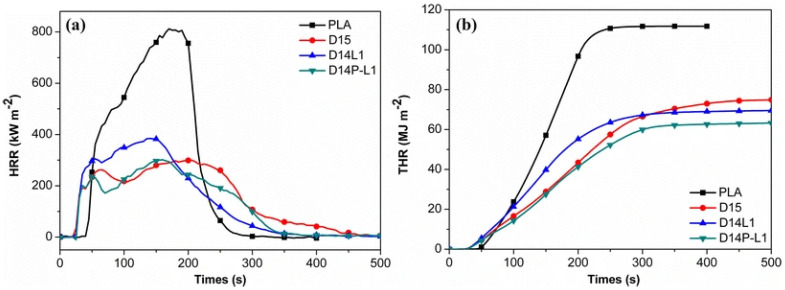
(**a**) HRR and (**b**) THR curves of PLA and PLA composites. Reproduced with permission for Jin et al. [83], Springer Nature.

**Figure 20 materials-16-06046-f020:**
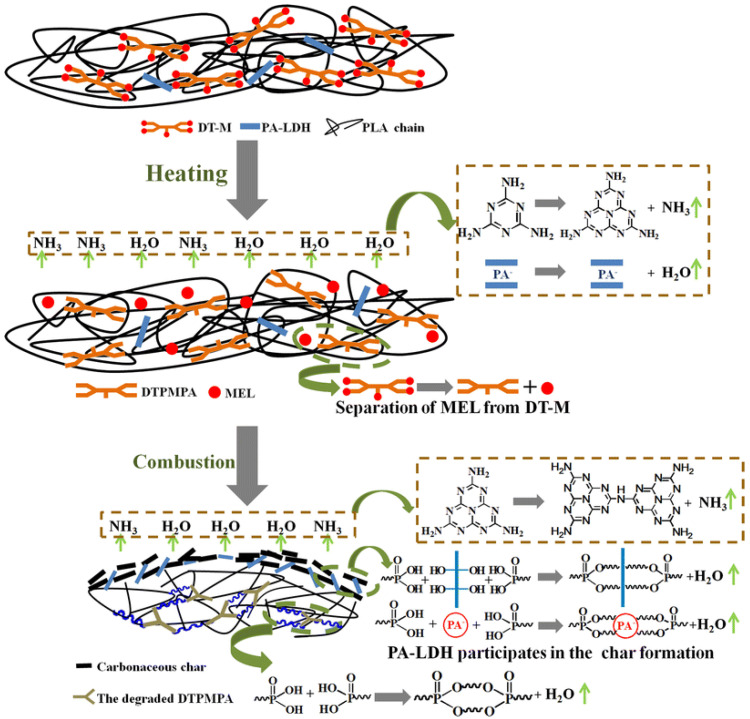
Possible FR mechanism of PLA/DT-M/PA–LDH composites [83]. Reproduced with permission for Jin et al. [83], Springer Nature.

**Table 1 materials-16-06046-t001:** A tabulated category of PLA feedstock.

Feedstock Class	Feedstock Source
1st generation	Potatoes, rice, wheat, sugar cane, corn, and sugar beet.
2nd generation	Switch grass, corn straw, palm fruit branches, sugar cane bagasse, wheat straw, and wood.
3rd generation	Algae-based biomass, food industry by-products, industrial waste, municipal waste.

**Table 2 materials-16-06046-t002:** Qualitative results and mean residual mass for PLA and composites obtained by vertical burning test UL94 V (standard deviation reported in parenthesis). Reproduced with permission from Fukushima et al. [33], Copyright 2010, Elsevier Science Ltd.

Sample	Flame to Holding Clamp	Flame Dripping	Ignited Cotton	Residual Mass (%)	Observations
PLA	No	Yes	Yes	64 (±4)	Intense dripping
PLA-3C30B	Yes	Yes	Yes	26 (±1)	Burning with drips, charring
PLA-3EG	Yes	Yes	Yes	31 (±5)	Burning with drips, charring
PLA-6EG	Yes	Yes	Yes	31 (±5)	
PLA-9EG	Yes	Yes	Yes	35 (±5)	
PLA–3C30B–3EG	Yes	No	Yes	40 (±2)	Compact char ^a^
PLA–3C30B–6EG	Yes	No	No	55 (±1)	Compact char
PLA–3C30B–9EG	Yes	No	No	58 (±5)	Compact char

^a^: The sample breaks into two pieces when the flame is up to the holding clamp. The fallen piece ignites cotton.

**Table 3 materials-16-06046-t003:** 2D NMTs-based FR and/or their amalgamates in PLA/PLA blends.

PLA Formulation	Processing Method	2D NMTs-Based FR	2D FR wt.%	2D FR Aspect Ratio	LOI %	UL94	Calorimetry Parameters (CCP)		TGA Data		Ref.
							pHRR Reduction %	OIT (°C)	*T*_5%_ (°C)	Residue %/Max. *T* (°C)	
		GR/graphene									
PLA/Ferric Phytate-graphene	MCD + CM	PLA/graphene	3	High	-	-	40	-	-	~5 (600 °C), N_2_	[69]
PLA/EG	MCD + CM	PLA/EG	12	High	-	HB	30	-	347	-(600 °C), air	[67]
PLA/APP/EG	MCD + CM	PLA/EG	11.25	-	36.5	V-0	38.3	-	322.9	9.3 (800 °C), N_2_	[68]
PLA/f-graphene	MCD + CM	PLA/graphene	3	High	-	-	40	-	-	~2 (600 °C), N_2_	[69]
		GO/GIO									
PLA/GO-3/4%CSFR	MCD + IM	PLA/GO	6.6	-	29.7	V-0	14.8	-	329.6	3.4 (800 °C), N_2_	[37]
PLA/polyphosphonate (BPPT) and polyethyleneimine-GO (M-GO) (PLA/2.4BPPT/0.6M-GO)	Solution mixing (SM)/casting + MC	PLA/GO	0.6	-	36	V-0	2.3	-	327	1.9 (600 °C), N_2_	[72]
PLA-4	SM/intercalation & film casting (FCT)	PLA/GPZ	2 of GPZ	-	27	V-2	39.4	-	-	3.5 (600 °C), N_2_	[71]
PLA/Co_3_O_4_/graphene	MCD	PLA/graphene	0.5	-	-	-	40	-	-	-	[50]
PLA/ZIF-8@GO, 0.5 wt.% (PLA-2)	SM/intercalation & film casting (FCT)	PLA/GO	0.5	High	24.0	V-2	-	-	-	-	[31]
PLA/graphene-hybrid (PLA/15%GOH)		PLA/graphene	15	-	-	V-0	47	-	-	-	[73]
		rGO									
rGO-AZOB (azo-boron (AZOB))/SMB (sodium metaborate (SMB))/PLA	SM + CM	PLA/rGO	0.5	-	31.2	V-0	76.5	-	265.3	9.81 (700 °C), N_2_	[65]
TRG/PLA	SM + CM	PLA/TrGO	10	-	-	-	-	197.9	274.0	44 (700 °C), N_2_	[5]
TRG/PLA/Py-PLA-OH	SM + CM	PLA/TrGO	10	-	-	-	-	204.2	304.7	39 (700 °C), N_2_	[5]
PLA-CO_3_O_4_/rGO	MCD	PLA/rGO	0.5–1	-	-	-	40	-	300 and 450	-	[50]
PLA/rGO	MCD + CM	PLA/rGO	2	-	-	-	14.9	-		41.7 (750 °C), N_2_	[74]
PLA/LDH-rGO (PLA-PPC/5d2D-Gc)	SM/casting + CM	PLA/rGO	2.5	-	-	-	50.24	-	-	-	[70]
PLA-PBS/5d2D-Gc	SM/casting + CM	PLA:PBS/rGO	2.5	-	-	-	29	-	-	-	[70]
		Clays									
PLA/aluminium diethylphosphinate (AlPi) O-MMT	MCD + CM	PLA/O-MMT	5	-	28	V-0	26.2	-	328	4.87 (700 °C) N_2_	[51]
PLA/O-MMT/FR (PLA/O-MMT/FR30)	MCD + CM	PLA/O-MMT	5	-	-	V-0	-	-	258.4	4.75 (700 °C) N_2_	[32]
PLA/O-MMT	SM + FCT	PLA/O-MMT	3	-	-	-	-	-	325	3.66 (800 °C) N_2_	[75]
PLA/O-MMT/bamboo fibres	MCD + CM	PLA/NC/recycles bamboo fibres	6	-	-	V-0	-	-	377.8	21.8 (850 °C) N_2_	[76]
PLA/C_30_B	MCD	PLA-Methyl-tallow-bis(2-hydroxyethyl)-ammonium treated MMT (2 wt.%)	4	-	-	-	38	-	-	-	[56]
PLA/O-MMT/Aluminum trihydrate	MCD	PLA/O-MMT	5	-	42	V-0	65	-	-	320 (650 °C), air	[77]
PLA/3C30B	MCD	PLA/CL	3 CL	200–1500		Limit			355	≥0.5 (500 °C), N_2_	[33]
PLA/3C30B/3EG	MCD	PLA/CL/EG	3 CL & EG	200–1500		HB			355	<10 (500 °C), N_2_	[33]
PLA/3C30B/6EG	MCD	PLA/CL/EG	3 CL & 6 EG	200–1500		HB			354	~10 (500 °C), N_2_	[33]
PLA/3C30B/9EG	MCD	PLA/CL/EG	3 CL & 9 EG	200–1500		HB			353	>11 (500 °C), N_2_	[33]
PLA/4 wt.% B104	MCD + melt spinning + knitting	PLA/CL	4	100–500	-	-	46	-	-	-	[24]
PLA/MWCNT-sepiolite NC (Sep)	MCD	PLA/MWCNT/Sep	2 & 10 (MWCNT & Sep)	100–300	-	-	45	-	334	11 (600 °C)	[78]
PLA/MWCNT-sepiolite NC-hemp fibres	MCD + CM	PLA/MWCNT/Sep-hemp fibres	20 (MWCNT & Sep) + 30 vol.% hemp	100–300	-	-	58	-	336	13 (600 °C), air	[78]
PLA/NC-Al diethylphosphinate (AlPi)	MCD + CM	PLA/NC	20	High	27.5	BC	36	-	333 for *T*_10%_	4.3 (600 °C)	[53]
PLA/NC	MCD + CM	PLA/NC	20	High	23	BC	51	-	333 for *T*_10%_	4.3 (600 °C)	[53]
PLA/Sepiolite/lignin (PLA-2)	MCD + IM	PLA/CL	3	High	-	-	37	-	310	16.4 (900 °C), N_2_	[79]
PLA/NC (PLA/15RC)	MCD + IM	PLA/NC	15	-	-	V-2	-	-	326	12.7 (750 °C), N_2_	[80]
P17M1C	3D printing	PLA/CL	1	-	27	V-0	42	-	-	5–10 (800 °C), N_2_	[49]
		MXenes									
PLA/MHCSN-Ti_3_C_2_T_x_ (MHCSN-TC)/SiAPP	MCD	PLA/MXene	1	-	32.7	V-0	64.1	-	-	-	[81]
PLA/benzyldimethylhexadecylammonium chloride (HDBAC) modified Ti_3_C_2_T_x_	MCD + CM	PLA/MXene	15	-	33.3	V-0	49.8%, 31.9%, 60.3% and 52.7%	-	327	10.33 (700 °C)	[60]
PLA/9,10-dihydro-9-oxa-10-phosphaphenanthrene-10-oxide (DOPO)- MXene (Ti_3_C_2_)	MCD + CM	PLA/MXene	3	High	33.7	V-0	-	-	-	-	[36]
PLA/IFR/MXene	MCD + CM	PLA/MXene	0.5	-	33.0	V-0	64.6	-	-	-	[63]
		LDH									
PLA	MCD	PLA	-	-	-	-	(pHRR: 480 W/g)	-	-	1.7 (700 °C)	[52]
PLA/FR-MgAl-LDH	MCD	PLA/LDH	2	-	-	-	46 (pHRR: 199 W/g)	-	-	1.7 (700 °C)	[52]
PLA/FR-ZnAl-LDH	MCD	PLA/LDH	2	-	-	-	58.5 (pHRR: 259 W/g)	-	-	58.5 (700 °C)	[52]
PLA/HPCP/LDH-SDS	MCD	PLA/LDH-SDS	2NiFe, 2NiAl, & 2NiCr	-	29	V-0	-	-	-	4.1, 6.3, & 7.2 (700 °C)	[82]
PLA/LDH-15% DT-M (PLA/DT-M/PA–LDH)	MCD + CM	PLA/LDH	1	-	38.9	V-0	63	-	350	10.6 (700 °C)	[83]
PLA/LDH (LCP-10)	Solution exfoliation + FCT	PLA/LDH	10	-	-	-	10	-	331.6	<10 (600 °C)	[84]
PLA/NiAl LDH/Silane coated APP, pentaerythritol phosphate (PEPA)	MCD	PLA/NiAl LDH	2.5	-	30	V-0	21	-	315	17.3 (700 °C)	[30]
MoS_2_											
PLA/MoS_2_/M _x_O_y_/CNTs	MCD	PLA/MoS_2_	~0.67	0.02–0.16	-	-	-	-	~323	5–10 (800 °C), N_2_	[34]

Footnote: Oxidative induction time (OIT); Burn-to-clamp (BC); Hexaphenoxycyclotriphosphazene (HPCP); sodium dodecyl sulfate (SDS); multi-walled carbon nanotubes (MWCNTs).

**Table 4 materials-16-06046-t004:** Potential/suggested niches of 2D NPs NP-reinforced PLA NCPs.

FR PLA-2D Material System	Electrical/Electronic	Automobile (AT)/Aerospace (AER)	Biomedical Materials	Furniture (FT)/Packaging (PK)/Building (BD) & Construction (CT)	Ref.
Graphene-based					
PLA/M-GO	✓	-	✓	✓	[72]
PLA/GO (PLA/GO-3/4%CSFR)	-	-	-	✓	[37]
PLA-GPZ	✓	-	-	-	[71]
PLA–C30B–EG	✓	✓	✓	✓	[33]
PLA/GO-3/4%CSFR	-	-	-	✓	[37]
PLA/15%GOH	-	-	✓	✓	[73]
Clay-based					
PLA/O-MMT	✓	-	-	-	[77]
P17M1C (PLA/MMT)	✓	✓	-	-	[49]
PLA/O-MMT	✓	✓	✓	✓	[53]
MXene-based					
PLA/MXene	✓	✓	✓	✓	[36]
PLA/MXene	✓	✓	✓	✓	[60]
LDH-based					
PLA/LDH	✓	✓	✓	✓	[52]
PLA/LDH	✓	✓	✓	✓	[84]
PLA/LDH	✓	✓	✓	✓	[82]
MoS_2_					
PLA/MoS_2_/M _x_O_y_/CNTs	✓	✓	-	-	[34]

Applicable (✓), Not applicable (-).

## Data Availability

Data are contained within the article.

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
