# Peer review of "Recent Advances and Outlook in 2D Nanomaterial-Based Flame-Retardant PLA Materials"

_materials, 2023, doi:10.3390/ma16176046_

Round 1

Reviewer 1 Report

Comment for materials-2511435 is listed as follows,

(1)   In title, please change the "2D nanomaterials-based poly (lactic acid) (PLA) flame-retardant materials: recent advances and outlook” into the “Recent Advances and Outlook in 2D Nanomaterials-based Flame-retardant PLA Materials”.

(2)   In section Keywords, please check the "Flame retarding materials", it did not been used in the paragraphs of manuscripts, e.g. change the “Flame retarding materials” into the “Flame-retardant”.

(3)   In section 1. Introduction, please check the name "Oliviera" in the “Alves de Oliviera et al. [19]” with the “19. Alves de Oliveira” in section References.

(4)   In page 12, please change the "Zhu et al. have emphasized ...[42] " into the " Zhu and Xanthos have emphasized ...[42] ", also check the "Bandypadhyay et al. [40]" with the “40. Bandyopadhyay, J., et al.” in section References.

(5)   In page 14, please check the subtitle no. "3.2. ". In page 15, please check the subtitle no. "3.1. ".

(6)   In Figure 12, Please check the "2020" in the figure name.

(7)   In Table 1, please add the second column name, e.g. "complexity".

(8)   Please change the equation number at the right sided end and in the paragraphs of manuscripts for all equations into the "(no.)", e.g. change the "(i)" into the "(1)", …etc.

None.

Author Response

Response to reviewers’ comments

The authors express their thanks to the editors and reviewers for their excellent comments/suggestions aimed at improving the quality of our manuscript. Please find below our responses to the comments/suggestions from the reviewers for your reference. We have made the changes throughout the revised manuscript highlighted in turquois color.

Reviewer # 1

Comment #1.   In title, please change the "2D nanomaterials-based poly (lactic acid) (PLA) flame-retardant materials: recent advances and outlook” into the “Recent Advances and Outlook in 2D Nanomaterials-based Flame-retardant PLA Materials”.

Response #1. Thank you for your observation: the title has been revised as per your suggestion.

Comment #2.   In section Keywords, please check the "Flame retarding materials", it did not been used in the paragraphs of manuscripts, e.g., change the “Flame retarding materials” into the “Flame-retardant”.

Response #2. The keyword has been revised as per your comment.

Comment #3. In section 1. Introduction, please check the name "Oliviera" in the “Alves de Oliviera et al. [19]” with the “19. Alves de Oliveira” in section References.

Response #3. The name has been revised accordingly.

Comment #4.   In page 12, please change the "Zhu et al. have emphasized ...[42] " into the " Zhu and Xanthos have emphasized ...[42] ", also check the "Bandypadhyay et al. [40]" with the “40. Bandyopadhyay, J., et al.” in section References.

Response #4. The name has been revised accordingly.

Comment #5. In page 14, please check the subtitle no. "3.2. ". In page 15, please check the subtitle no. "3.1. ".

Response #5. As per your suggestive comments, we have revised the sections accordingly.

Comment #6. In Figure 12, Please check the "2020" in the figure name.

Response #6. We appreciate your valued comments: we have corrected the year as per your comment.

Comment #7. In Table 1, please add the second column name, e.g. "e.g.,lexity".

Response #7. The second column name has been added as per your comments.

Comment #8.   Please change the equation number at the right sided end and in the paragraphs of manuscripts for all equations into the "(no.)", e.g. ce.g.,e the "(i)" into the "(1)", …etc.

Response #8. The numbering of the equations has been revised as per your comments.

Reviewer 2 Report

The aim of the paper is to review the last decades litterature related to the 2D flame readants for PLA; it is notheworty that the topic is interesting and also deeply studied.

Anyway there are many flaws that limit the redibility of the paper; i have listed a few below and authors can find more in the attached file:

1) Acronymus should be explicited the first time the word is used: for example, the first time authors use FR (in the abstract) the acronymus is not explicited;

2) Introduction is too long;

3) the paragraph "PLA feedstock" is too long: because the paper review the FR for PLA, readers don' need a large overview regarding the PLA sources;

4) the PLA synthesis paragraph is too similar to the paper:

DOI: 10.3389/fbioe.2019.00259

Please modify in order to delete the similitudes; moreover the section is too long;

5) I think it is not appropriate to slipt the Processing paragraph in sub-paragraphs: create a single paragraph where authors describe the single techniques;

6) Although the number of references is adequate, it seems that the main section of the paper is simple list of scientific results: the paper quality could be really improved if authors could rearrange that results highlighting different filler aspects, such as effect of fillers dispersion grade on FR, effect of filler functionalities on FR and so on;

The paper could be published but is should be improved considering these aspects together with those reported in the attached file; moreover i suggest to carefully check english style and tipyng errors.

English style should be improved.

Author Response

Reviewer # 2

The aim of the paper is to review the last decades litterature related to the 2D flame readants for PLA; it is notheworty that the topic is interesting and also deeply studied.
Anyway there are many flaws that limit the redibility of the paper; i have listed a few below and authors can find more in the attached file:

Comment #1. Acronymus should be explicited the first time the word is used: for example, the first time authors use FR (in the abstract) the acronymus is not explicited;
Response #1. We appreciate your observation and comments in this regard: the abbreviations have been revised as per your comment.

Comment #2. Introduction is too long;

Response #2. We have revised the introduction as per your comment.

Comment #3. the paragraph "PLA feedstock" is too long: because the paper review the FR for PLA, readers don' need a large overview regarding the PLA sources;

Response #3. We have revised the paragraph as per your comment.

Comment #4. the PLA synthesis paragraph is too similar to the paper:

DOI: 10.3389/fbioe.2019.00259. Please modify in order to delete the similitudes; moreover the section is too long;

Response #4. This section has been revised as per your suggestion/comment.

Comment #5. I think it is not appropriate to slipt the Processing paragraph in sub-paragraphs: create a single paragraph where authors describe the single techniques;

Response #5. We have revised the processing portion as a stand-alone section.

Comment #6. Although the number of references is adequate, it seems that the main section of the paper is simple list of scientific results: the paper quality could be really improved if authors could rearrange that results highlighting different filler aspects, such as effect of fillers dispersion grade on FR, effect of filler functionalities on FR and so on;

Response #6. As per your valuable comment/suggestion, a column has been added in Table 3 showing the aspect ratio of the utilized 2D NPs used for easy correlation of with the flame retardancy of their host matrix (PLA and/or its blends). Though most of the literatures failed to mention the aspect ratio of the 2D NPs utilized in their studies.

Comment #7. The paper could be published but is should be improved considering these aspects together with those reported in the attached file; moreover i suggest to carefully check english style and tipyng errors.

Response #7. The English style has been improved as per your comments.

Comment #8. English style should be improved.

Response #8. The English style has been improved as per your comments.

Reviewer 3 Report

It is a very interesting and desirable publication, which will provide authors with many citations. Nevertheless, minor technical corrections are required:
1. In Figure 4 - markings on the x, y, and z axes are hard to read. Please correct it
2. On page 15 we have fonts of different sizes.
3. Improve the quality of Figures 11, 12, 16, 17, 18.

Author Response

Reviewer # 3

It is a very interesting and desirable publication, which will provide authors with many citations. Nevertheless, minor technical corrections are required:
Comment #1. In Figure 4 - markings on the x, y, and z axes are hard to read. Please correct it.

Response #1. Your suggestions/comments are well appreciated: the figure has been revised accordingly.

  1. On page 15 we have fonts of different sizes.

Response #2. The font size difference have been rectified as per your comment.

  1. Improve the quality of Figures 11, 12, 16, 17, 18.

Response #3. The quality of the figures used here is the same as in the original manuscripts.

Reviewer 4 Report

The manuscript titled "2D Nanomaterials-based Poly (lactic acid) (PLA) Flame-retardant Materials: Recent Advances and Outlook" presents a promising review of recent advancements in flame-retardants for PLA, with a specific focus on 2D nanosystems and their composites. The abstract effectively sets the stage for the article, conveying the importance of finding ecologically friendly flame-retardant solutions for PLA. The conclusion provides an informative overview of the properties and applications of 2D NMTs, although a stronger connection to PLA flame-retardant materials could be made.

The abstract could benefit from including more specific details about the recent advancements and breakthroughs in 2D nanosystems-based flame-retardants for PLA. Mentioning a few key examples or specific findings could help generate more interest from readers.

While the abstract provides an overview of the association between flame-retardant loadings and efficiency for different FR-PLA systems, briefly mentioning some key results or trends in this regard might be helpful.

The conclusion could be strengthened by directly linking the discussed applications of 2D NMTs, including flame-retardants, with the specific context of PLA nanocomposite systems. This would reinforce the relevance of the discussed 2D NMTs to the main focus of the manuscript.

While the conclusion highlights the exceptional properties of 2D NMTs, it might be helpful to briefly mention the challenges or limitations associated with their integration into PLA nanocomposite systems for flame-retardant applications.

Overall, the manuscript holds significant potential and relevance for readers interested in the intersection of nanotechnology, flame-retardant materials, and PLA. To enhance the manuscript, the authors may consider expanding on the specific recent advancements and key findings in the abstract and directly linking the properties and applications of 2D NMTs to PLA nanocomposite systems in the conclusion. Additionally, addressing any potential limitations or challenges associated with the integration of 2D NMTs in PLA flame-retardant materials would further strengthen the manuscript.

Based on the provided manuscript, the quality of the English language in the manuscript appears to be moderate. Therefore, moderate editing of the English language is required.

Author Response

Reviewer # 4

Comment #1. The manuscript titled "2D Nanomaterials-based Poly (lactic acid) (PLA) Flame-retardant Materials: Recent Advances and Outlook" presents a promising review of recent advancements in flame-retardants for PLA, with a specific focus on 2D nanosystems and their composites. The abstract effectively sets the stage for the article, conveying the importance of finding ecologically friendly flame-retardant solutions for PLA. The conclusion provides an informative overview of the properties and applications of 2D NMTs, although a stronger connection to PLA flame-retardant materials could be made.

Response #1. We appreciate your valuable feedback and suggestions/comments: we have revised the manuscript as per your comments.

Comment #2. The abstract could benefit from including more specific details about the recent advancements and breakthroughs in 2D nanosystems-based flame-retardants for PLA. Mentioning a few key examples or specific findings could help generate more interest from readers.

Response #2. As per your comment, we have added the findings as per literature in the abstract.

Comment #3. While the abstract provides an overview of the association between flame-retardant loadings and efficiency for different FR-PLA systems, briefly mentioning some key results or trends in this regard might be helpful.

Response #3. We have revised the abstract accordingly as per your suggestions.

Comment #4. The conclusion could be strengthened by directly linking the discussed applications of 2D NMTs, including flame-retardants, with the specific context of PLA nanocomposite systems. This would reinforce the relevance of the discussed 2D NMTs to the main focus of the manuscript.

Response #4. As per your suggestive comment, we have revised the conclusion accordingly.

Comment #5. While the conclusion highlights the exceptional properties of 2D NMTs, it might be helpful to briefly mention the challenges or limitations associated with their integration into PLA nanocomposite systems for flame-retardant applications.

Response #5. As per your suggestive comment, we have revised the conclusion accordingly.

Comment #6. Overall, the manuscript holds significant potential and relevance for readers interested in the intersection of nanotechnology, flame-retardant materials, and PLA. To enhance the manuscript, the authors may consider expanding on the specific recent advancements and key findings in the abstract and directly linking the properties and applications of 2D NMTs to PLA nanocomposite systems in the conclusion. Additionally, addressing any potential limitations or challenges associated with the integration of 2D NMTs in PLA flame-retardant materials would further strengthen the manuscript.

Response #6. As per your suggestive comment, we have revised the abstract and conclusion accordingly.

Comment #7. Based on the provided manuscript, the quality of the English language in the manuscript appears to be moderate. Therefore, moderate editing of the English language is required.

Response #7. As per your suggestive comment, we have revised the English language the revised manuscript.

Round 2

Reviewer 1 Report

Accept.

None.

Author Response

we thoroughly proof-read our manuscript

Reviewer 4 Report

I encourage you to review your self-citations and prioritize external sources that strengthen your arguments and provide a broader context. This will help ensure that your manuscript maintains scholarly rigour and integrity.

Certain sections of the manuscript exhibit language issues that can occasionally impede the clarity and flow of your ideas. To enhance your work's overall readability and professionalism, I recommend a thorough review to address grammatical errors, sentence structure, and word choice.

Author Response

most of them are linked with the content described in this work. However, we implemented your suggestions, including thoroughly proof-read our manuscript